# Specific Design Approach of Croatian Architect Dinko Kovačić: The Coexistence of Modernism and Tradition in the Second Half of the 20th Century

**Vesna Perković Jović [1,*] and Neda Mrinjek Kliska [2,*]**

1 Department of Buildings, Faculty of Civil Engineering, Architecture and Geodesy, University of Split, HR-21000 Split, Croatia
2 Department of Architectural Technology and Building Science, Faculty of Architecture, University of Zagreb, HR-10000 Zagreb, Croatia
* Correspondence: vesna.perkovic@gradst.hr (V.P.J.); nmrinjek@arhitekt.hr (N.M.K.)

**Abstract:** Dinko Kovačić is a prominent Croatian architect and university professor. His design approach is characterised by exceptional empathy that results in architectural works of intense connection with the environment as well as with that of their future users. Although many of Kovačić's works have been published in the daily press and professional publications, the complete oeuvre of this architect has so far not been the subject of scientific research. The aim of this scientific work is to look at his specific design approach based on the analysis of representative examples. Research methods in this paper include the analysis of primary and secondary sources and on-site observation. The article gives a systematic presentation of Dinko Kovačić's work as related to his specific approach, which integrates the modernism of the second half of the 20th century and the Mediterranean tradition.

**Keywords:** architect Dinko Kovačić; second half of the 20th century; residential building; family house; school; hotel; modernism and tradition; Dalmatia; Croatia; Mediterranean

> "I have searched for an understanding and measure. An understanding between modernity and tradition. A measure between satisfaction and happiness." (Dinko Kovačić)

## 1. Introduction

Dinko Kovačić (Split, 1938) is a prominent Croatian architect who worked in the second half of the 20th and the first decade of the 21st century. Given the fact that he has lived and worked in Split all his life, most of the buildings he has designed are in Split and its surrounding area. Only towards the end of his career were his buildings executed in other parts of Croatia.

Croatian modern architecture is the focus of numerous recent professional and scientific studies. Monographs have been written about many Croatian architects, for example, Branko Kincl, Marijan Hržić, Miroslav Begović, Lavoslav Horvat, Zoja Dumengjić, Frano Gotovac, and Nikola Filipović [1–7].

The work of architect Dinko Kovačić is recognised as extremely valuable. In most encyclopaedic and lexicographic editions concerning culture and art there is a note or a short entry on Dinko Kovačić and his work [8–13]. Many authors dealing with the architecture of the post-war period have written professional reviews about his realisations. They discussed the specific typology of construction, the stylistic direction, and the particular area of construction, etc. [14–19]. However, for a more complete review of his work, one must turn to the exhibition catalogues, which the architect prepared himself [20–22].

In contrast, numerous articles were written in the local daily press about Kovačić's realisations. His houses are eloquent, appealing, picturesque, photogenic, innovative,

and rich with details. Architects, art historians, and journalists liked to write about them. Kovačić gladly talked with journalists about his current work. He always tried to explain the idea of his project in simple words, to present some interesting details or an anecdote about a particular building. This ease of storytelling is one of the reasons there are more than 200 articles about Kovačić's work in the daily press and in professional publications. Apart from the construction of buildings, the articles covered every important activity of the architect, such as his exhibitions and summer workshops on the Island of Brač with the students of the Faculty of Architecture in Zagreb. The articles also convey his opinions on certain important topics related to his profession, on topics concerning his city, culture, and so on [23–84]. Nonetheless, Dinko Kovačić's rich oeuvre has not been fully or comprehensively presented in any scientific article until now.

His work was presented at several large exhibitions held in Croatia (in Zagreb in 2000 and 2014 and in Split in 2000/2001 and 2017) and abroad, in Paris, 2002, Brussels, 2002/2003, Strasbourg, 2003, and Ljubljana (Slovenia), 2013 [85–103]. The apartment block in Ljubićeva Street, the work of architects Dinko Kovačić and Mihajlo Zorić, was presented at the exhibition "Toward a Concrete Utopia: Archi-tecture in Yugoslavia 1948–1980", organised by Martino Stierli at the Museum of Modern Art (MoMA) in New York (2018–2019) [102–108].

Dinko Kovačić has won many professional awards for his realisations. The most prominent being the three professional awards for the residential complex on Šime Ljubića Street—the "Borba" Federal Award, the Award of the 8th Zagreb Salon, and the "Vladimir Nazor" Annual Award, in 1974 (with M. Zorić)—and the "Drago Galić" Annual Award in 2011 for the Stupalo house [109–118]. He received several prestigious lifetime achievement awards: the "Vladimir Nazor" Award in 2011 given by the Ministry of Culture and Media, the "Slobodna Dalmacija" Award in 2018, awarded by the newspaper of the same name, and the "Viktor Kovačić" Award in 2019 awarded by the Croatian Chamber of Architects [119–121]. For his noteworthy activity in the field of architecture, Dinko Kovačić has been a member of HAZU (Croatian Academy of Sciences and Arts), the Department of Fine Arts, since 2006. As stated in the explanation upon acceptance: "With numerous designs of multi-storey residential buildings, school and faculty buildings, hotels and business facilities, architect Dinko Kovačić demonstrated exceptional graphic virtuosity, wealth of spatial composition and creative interpretation of progressive architectural morphology with a subtle feeling for local ambient tradition" [122].

The aim of this article is to present the design method of Kovačić's work. From his rich body of professional work, which he focused on continuously for about 50 years, the most striking buildings have been chosen to illustrate his creativity. The architect's specific method of design is presented in his words cited in many articles. His assumed design method is verified via the examination of his most prominent buildings. The contribution of this article is to present Kovačić's work and his design process to the international public.

## 2. Architect Dinko Kovačić's Professional Background

Architect Dinko Kovačić was born in Split on September 30, 1938. He attended "Ćiro Gamulin" Realschule in his hometown and graduated in 1957. After graduation, he enrolled at the Faculty of Architecture in Sarajevo, but continued his studies in Zagreb after the first year. He graduated from the Faculty of Architecture in Zagreb in 1963 under the mentorship of Professor Vladimir Turina with the topic "Tourist Settlement in Zablače near Šibenik".

After graduating, he returned to his native Split, where he was employed by the "Tehnogradnja" Construction Company. In 1966, he moved to the "I. L. Lavčević" Construction Company. From 1979, he worked at the Institute of Architecture at the Faculty of Architecture in Zagreb. He was professor at the Faculty of Architecture in Zagreb from 1994 until his retirement in 2008, teaching courses in Buildings for Tourism and Leisure and Integral Design Studio. Parallel to his teaching activities, Kovačić designed very intensively, running his own office and, at the same time, completing some important projects. He lived between Split and Zagreb, organised exhibitions of his designs, both in the country

and abroad, and held lectures on his works and other architectural topics. Nevertheless, he remained attached to his city throughout his entire professional life.

Dinko Kovačić spent a large part of his professional life working within the framework of the planned economy in former Yugoslavia, a socialist country. It was only from 1991, with the declaration of independence of Croatia, that the transition towards a market economy began. These socio-political circumstances significantly affected the architectural profession. After the Second World War, the state was the only investor, and architects practised their profession in the project offices of large construction companies or in the statal institutes. Shortly after the Second World War, the largest proportion of construction projects were related to residential construction due to the rapid urbanization and industrialization of the country. In parallel, educational buildings were built to increase the educated population. From the 1960s, Yugoslavia started to open to Western countries and develop tourism, and intensive construction of tourist facilities began. With the oil crisis of the 1980s, the intensity of construction decreased and almost completely stopped with the Homeland War. After the death of President Josip Broz Tito in 1980, the period of political instability started in former Yugoslavia, culminating in the Homeland War. In 1991, Croatia declared independence from Yugoslavia. Since then, the architectural profession has been governed by new socio-economic circumstances, for example, being almost completely deregulated and having to adapt to neoliberal capitalism.

However, besides the conditions of socialist and neoliberal society, there is the poetry of real life. Dinko Kovačić has lived all his life in a traditional house on the peripheral part of Veli Varoš, a common suburb at the foot of Marjan Hill, on the western side of the historic centre of Split and Diocletian's Palace. The Varoš area is rich in greenery that descends from Marjan to the seafront. Traditional stone houses, narrow stone-paved streets, and a few small gardens create the typical atmosphere of a densely built Mediterranean settlement on a hillside. The Kovačić house is accessed from the terrace on the promenade connecting *lungomare* with Marjan hilltop. Because he lived in this exceptional place all his life, Kovačić wanted to give other people an architectural space that is a starting point for happy living.

Kovačić has been a birder all his life. He has always had a large birdcage on the terrace of his house. In his office at Bosanska Street in Diocletian's Palace, birdhouses were hung on the old stone walls. Observing the birds building their nests, Kovačić noticed the rationality and sense of measure in nature and life and adopted it in his architecture: "The swallow builds the nest in the proportions of its body so that it holds heat, even though there is plenty of mud and twigs it uses to build the nest. Measure is the most important thing, not only in architecture but also in life" [33] (p. 276). "Birds are", he would say, "beings without vanity. Vanity is the man's greatest burden which makes him constantly and needlessly want more and more, bigger rooms, terraces, windows..." "Measuring the measure is the duty and the rule of our profession" [56] (p. 20). As Kovačić points out, the nest is a symbol of construction. A picture of a swallow's nest featured on the poster of his exhibition in Brussels in 2002/2003, and on the poster announcing his lecture "*Split 3—my memories*" in 2018 in Split. On the front door of his office, the same picture is standing next to the inscription "Architect Dinko Kovačić's Office".

Architect Kovačić is recognised as an excellent designer of various architectural types. His oeuvre is dominated by residential architecture starting with the large residential ensembles of Split 3 from the 1970s, through to a series of residential buildings in Split, Rovinj, and Supetar, to family houses. He is also renowned for holiday houses, which he designed for his friends in parallel with his "serious" tasks during his entire professional practice. Kovačić realised several school buildings and hotels, such as the Žrnovnica Primary School from 1990, the Secondary School Centre from 1992, the Faculty of Economics built in Split in two phases, in 2001 and 2006, and his last finished building, the Meterize Elementary School in Šibenik from 2010–2012. Among the hotels, two projects stand out: Hotel "Bretanide" in Bol on the Island of Brač from 1985 and the "Uvala" Hotel in Lapad

Bay in Dubrovnik from 2003, a very successful project of reconstruction and adaptation of the existing resort from the period of socialism.

Because of his continuous and fruitful professional activity, he is one of the most respected professors of the Faculty of Architecture. In his 2016 monograph on academic Branko Kincl, architect Zlatko Karač distinguishes Dinko Kovačić, along with some other professors of the Zagreb Faculty of Architecture, for having academic authority not only due to the quality of his work but also because of the quantity of his experience. "These are all professors who not only formally taught architecture, but have also created outstanding works of their time, confirmed on numerous 'construction sites'!" [1] (p. 83).

Dinko Kovačić taught students the skill of designing based on his own experience. He always emphasised: "Every project must have three elements, or phases: knowledge, conviction, and enthusiasm. Without enthusiasm, it makes no sense to even pick up a pencil" [66].

He conceived the Summer School of Architecture in Bol on Brač called "Modernity and tradition or... as written in stone" and successfully led it from 1997 until his retirement. Students and lecturers talked and debated about architecture, its role in society, and its relationship with natural and architectural heritage. Academician Jakša Fiamengo reported on the Summer School in the daily press. The titles of articles in "Slobodna Dalmacija" best describe the atmosphere of the school: "The light of the Brač stone", "Architects are producers of happiness", and "Architectural school of wisdom" [123–125].

Dinko Kovačić belongs to the generation that began its professional journey at a time when many of the shortcomings of the International style, which prevailed in post-war Europe, were becoming clearly visible. These shortcomings were partly related to the narrow interpretation of functionalism and the reductivism of modernism, and partly to the uncritical application of new building technologies. After the Second World War, at first, due to the need to rebuild war-torn Europe, and then due to social changes brought about by rapid industrialisation, deagrarianisation, and urbanisation, there was a need for fast construction and the expansion of cities. Very similar types of prefabricated construction were used across Europe, which further impoverished the reduced architectural language of modernism by their technological and economic requirements. On the other hand, it became clear that the functionalist approach to design imposed certain patterns on space use, which did not correspond to the users' real needs and habits.

The first research using a participatory approach in design appeared at the beginning of the 1960s. As a result, users were given a greater opportunity to influence the future space within the scope of their jurisdiction. Although participatory design did not take off to a great extent, it clearly addressed the shortcomings of strict functionalism and, in this sense, influenced the future generations of architects.

Already in the 1950s, there was a stronger awareness of the importance of the unity of experience and material in architecture. This was manifested in a series of seminal works in the 1960s that criticised the degradation of the built environment because of economic and technological pressure, which ignored the people's need for identity and connection with place. The importance of the role of images and symbols in architecture was becoming stronger, as was the connection with the historical context. The return to pre-industrial urban forms became an amalgam for an impoverished architectural language and devastated environment. The exact reuse of the historical elements, which resulted in a scenographic, nontectonic approach, as well as the superficial manipulation of signs, emerged as an essential feature of postmodernist architecture.

Simultaneously, some architects in different parts of the world managed to establish a balance between the social and psychological meaning and the tectonic nature of the architectural work. The reinterpretation of the cultural context expressed through modern technological means was recognised as a common thread of critical regionalism [126–128].

Such an approach in design, which accepts and affirms the achievements of universal civilisation while representing the values of a specific culture, was the basis of the architect Dinko Kovačić's specific work. His thinking, proposed concepts, and realisations place him

next to his European counterparts. Kovačić relied on tradition and transformed its apparent forms into a modern repertoire of design tools with which he shaped his buildings. These elements are very recognisable and bear the architect's strong personal touch.

Kovačić was intrigued by the problem of alienation in modern society. He strived to offer people a pleasant place to live in large apartment blocks and towers. The educational buildings—schools and colleges—designed by Kovačić have become places not only for studying, but also places for socialising and spending time due to their spatial organisation, which follows the pattern of a Mediterranean city. While designing hotels, he followed the same spatial pattern, enriching it with many unexpected details striving to connect the tourist with a place. His design approach is characterised by exceptional empathy, which resulted in architectural works of intense connection with the environment as well as with that of their future users.

### 3. Motive for the Research and Methodology

The consistency of style in the architectural work of Dinko Kovačić and the perseverance of the architect in implementing his idea, as well as a confident and argumentative attitude, motivated this research. We aimed to highlight the specific design method that forms the fabric of Kovačić's work. To investigate this work as comprehensively as possible, the architect was recently interviewed on several occasions, starting from 2018. Documentation from the architect's archive—the execution drawings of buildings, the architect's drawings and sketches, and the competition designs that the architect kept in his archive—were all analysed. Over the past years, all the buildings were analysed in situ. Representative examples of buildings that best portray his oeuvre are presented in this paper. In addition, numerous articles written by various authors in the daily and professional press about Kovačić's buildings were reviewed. The interviews with Kovačić in which the architect talked about his design method or about his achievements from his point of view proved to be particularly valuable. The aim of this paper is to highlight the most important determinants of Kovačić's life-long architectural work.

### 4. Analysis of Selected Buildings by Architect Dinko Kovačić

The analysed buildings of the architect Dinko Kovačić will enable an overview of his opus in the visual-formative, programmatic, and typological sense. Buildings no. 4.1–4.5. are in Dalmatia, in Split-Dalmatia County, which occupies the central part of the Croatian coast on the Adriatic Sea. Analysed building no. 4.6 is in Rijeka in Primorje-Gorski Kotar County. The buildings are arranged in chronological order: 4.1—Ljubićeva Street in Split 3 from 1972/1973; 4.2—Marjanović-Popović semi-detached house in Meje, Split from 1972; 4.3—Hotel "Bretanide" in Bol on Brač built in two phases, in 1984 and 2006; 4.4—Secondary School Centre in Split 3 from 1989 to 1992; 4.5—Stupalo family house in Meje, Split from 2000; and 4.6—Ceremonial object of the Drenova cemetery in Rijeka from 2005–2007 (Figures 1 and 2).

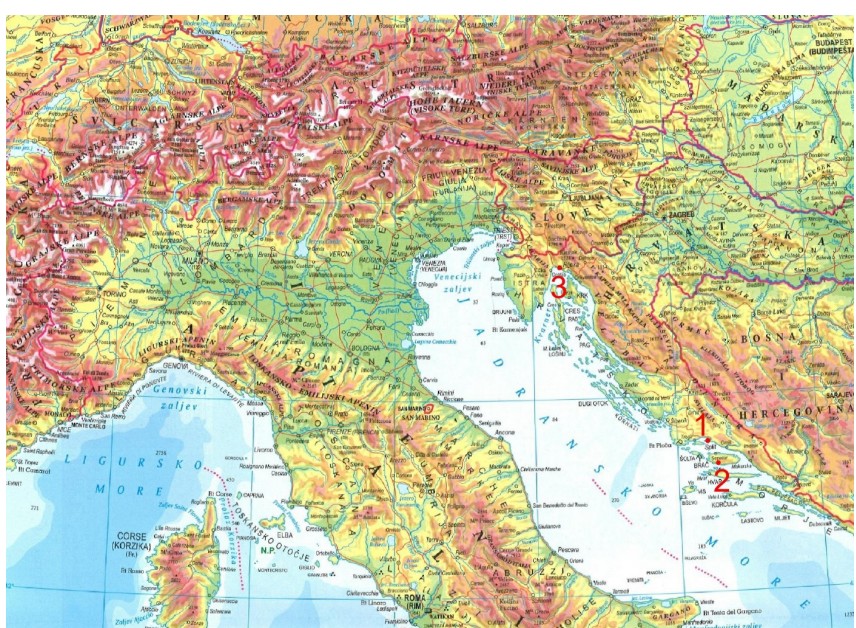

**Figure 1.** Position of the Republic of Croatia in the south of Europe. Marks on the map 1—City of Split; 2—Municipality of Bol, the island of Brač; and 3—City of Rijeka. Source: Geographical *atlas: for high schools and vocational schools*. Školska knjiga: Zagreb, Croatia, 2021, p. 36. Marks 1–3 added by the author of the article.

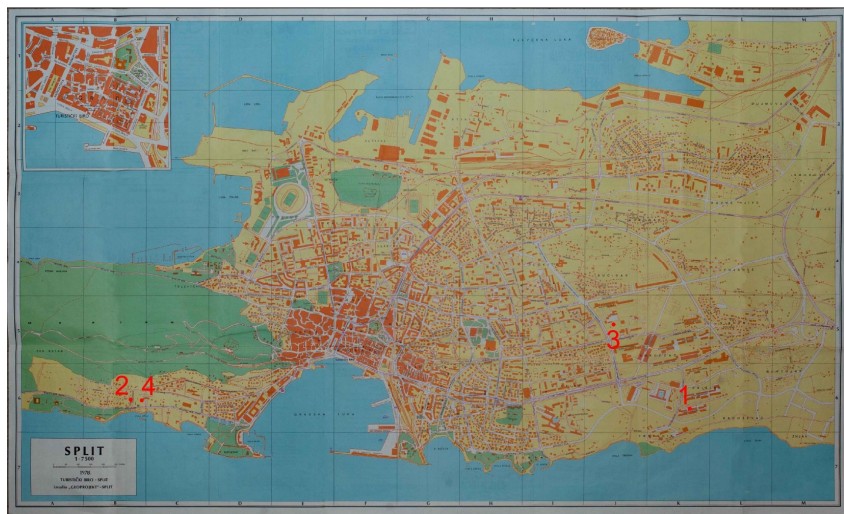

**Figure 2.** Plan of the city of Split, 1978. Kovačić's buildings located in Split and analysed in this article are marked on the map with numbers: 1—Residential buildings in Ljubićeva Street in Split 3; 2—Marjanović-Popović semi-detached house in Meje; 3—Secondary School Centre in Split 3; and 4—Stupalo family house in Meje. Source: Author's archive. Marks 1–4 added by the author of the article.

### 4.1. Ljubićeva Street in Split 3

Architect Dinko Kovačić started his career at the time of the great housing crisis in Split and in all major cities of the former Yugoslavia. During the late 1950s and early 1960s, many new residents from the nearby regions of Dalmatian Zagora and the islands came to Split for employment, as did military personnel, both active and retired, from distant parts of Yugoslavia. The municipality of Split was supposed to provide housing for these new residents. The city began to expand towards the periphery and the Split Field, starting with substantial construction work. Most of the residential buildings that were built did not stand out in terms of more ambitious approaches in the design of the apartment space

or the external appearance of the building [17] (p. 163), [129] (pp. 82–104, 124–135). The offered designs were often standardised to achieve the expected results in the shortest possible timeframe. However, among the large number of apartments built, there were also some high-quality projects.

Project programs that prescribed the content and the area of the apartment, and the size of and equipment in individual rooms were mandatory. Some designers were not satisfied with this fact, considering that it violated the architects' freedom of expression [130]. Yet, from a present-day point of view, the urban design of residential areas often surpasses the quality of housing construction in Split from the period after the Homeland War. The abovementioned settlements were enriched with parks with kindergartens, schools, and shops—contents that have been neglected in areas built since the change in socio-economic relations in the mid-1990s.

The economy of construction in residential buildings was of the utmost importance. The abovementioned programs tried to contribute to the rationalisation and standardisation of the design process and execution, which led to the greatest possible cost-effectiveness. The requirements of economisation and standardisation were established as early as 1928 as the basis of modern architecture at the International Congress of Modern Architecture—CIAM in La Sarraz [126] (p. 293). After the Second World War, in the newly founded state Yugoslavia modern architecture, rationalism and standardisation were encouraged to speed-up construction. Nevertheless, this approach was also considered progressive and appropriate to the new economic and political system—socialism. That was particularly the case in residential building, leaving the model of bourgeois rental apartments behind.

Kovačić's first job, after graduation in 1963, was at the project office within the "Tehno-gradnja" Construction Company, which was the most common form of architectural practice in the post-war period. His first project was the design of buildings for a residential complex in the Lokve area in Split, for the city competition, which was announced in 1966. At the competition, he won first prize and, following his proposal, the settlement was built in 1966–1968. In the early 1970s, as an already successful young designer, he assumed the position of designer in the project office within the "I. L. Lavčević" Construction Company from Split. This was at the time when the development of the new city area of Split 3 began.

The competition for the urban design of Split 3 was announced in 1968 at the federal level of former Yugoslavia. It was necessary to urbanise a large undeveloped area on the eastern part of the Split peninsula where 50,000 inhabitants were to live. Among 18 submitted works, first prize was won by urban planners Vladimir Braco Mušič, Marjan Bežan, and Nives Starc from the Urban Planning Institute of Slovenia [131] (p. 25). The basic idea of the winning entry was the return to the streets and to the traditional life in the Mediterranean. The architects came up with a system of parallel pedestrian streets in the east–west direction with which, as Vladimir Mušič points out, they tried to "contribute to a way out from the crisis of modern urbanism of scattered 'building spots' in the greenery, as well as parcelled out interactions" [132] (p. 8). The topic was very contemporary; similar to what was being studied by many European architects, for example, Giorgio Grassi, Aldo Rossi, and Vittorio Gregotti [126] (pp. 318–319). High-rise buildings were placed on the north side and low-rise buildings on the south side of the street. This allowed most of the apartments to enjoy the sun light and a view of the sea (Figure 3).

Split 3 represents a major leap in development because the construction companies from Split united in the so-called Joint Construction Operations of Split with the goal of more economical construction. Until then, Split's residential areas were built on smaller areas that were within reach of the arm of a crane, which is why they were called crane settlements.

The designing, financing, and development of Split 3 was organised by architect Josip Vojnović from the Company for the Construction of Split.

Shortly after the urban design competition, another competition for the designers of individual streets in Split 3 was launched. This was a competition at the city level, among designers from architectural offices in Split, although a competition at the federal level was also considered. The competition determined the future designers of the buildings of Split

3, instead of the concept designs of the buildings as was usual. The selected architects of the future buildings and streets of Split 3 were Dinko Kovačić and Mihajlo Zorić, Frano Gotovac, Danko Lendić, Marjan Cerar, and Ivo Radić.

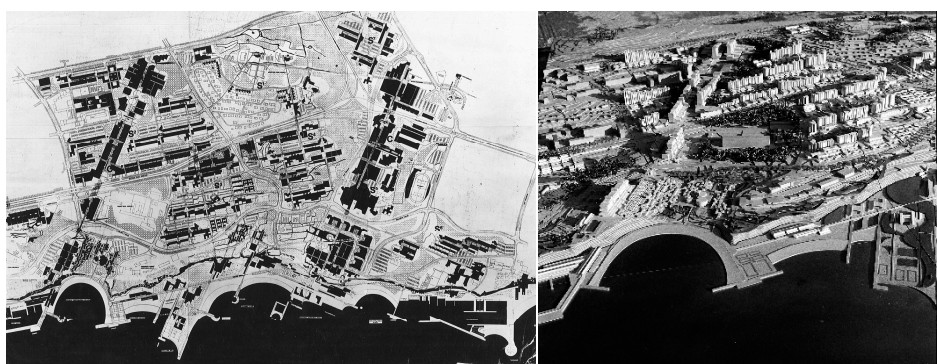

**Figure 3.** Urban project of Split 3 (**left**) and its model (**right**). Source: Author's archive.

In 1972/1973, architects Dinko Kovačić and Mihajlo Zorić designed an innovative project in the southern part of Split 3—Trstenik, in Ljubićeva Street—the street closest to the sea front (anagraphic designation: 1–21 Šime Ljubića Street, Split, former Borozan Brothers' Street—high-rise buildings with odd and low-rise buildings with even street's numbers).

According to Kovačić, the projects of three 16-storey residential towers (anagraphic designation: 14, 16 Alojzija Stepinca Street; 7 Šižgorićeva Street, formerly Paićeva Street) built in 1966 in the nearby Lokve area paved the way for the concept design of Ljubićeva Street [Interview with architect Dinko Kovačić, done in Split from 2018 up to 2023.]. These were the first buildings he designed independently after finishing his studies in Zagreb and beginning to work at the "Tehnogradnja" Construction Company. He won the city competition for the design of towers and the neighbouring multi-storey residential building in Lokve (anagraphic designation: 2–10 Alojzija Stepinca Steet, Split, formerly Paićeva Street). The towers with a square floor plan are formed by alternating sequences of solid-wall verticals and recessed strips with French windows along the entire height of the building. He designed four identical three-room apartments in the towers. When designing apartments, he took care over the details that met human needs and make staying in them more pleasant. Kovačić envisaged flowers in shallow loggias in front of the French windows becoming an extension of the interior.

Kovačić described the building's emergence as the growth of "concrete horticulture". The building, like a plant that is freed from the vertical dimension, "grows out of the ground (without the sidewalks and plinths) and its end does not have a cornice, so it does not mean completion in the architectural sense". The height of a skyscraper is defined by the planning guidelines within the urban development plans and the financial capacity of the investors [73] (Figure 4).

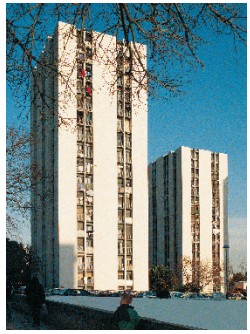 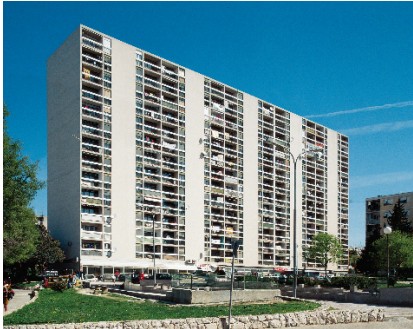

**Figure 4.** Skyscrapers (**left**) and residential building (**right**) in Lokve district in Split. Source: Dinko Kovačić's archive, unknown author.

It is indisputable that the skyscrapers in Lokve and the residential streets in Split 3 are conceptually related. The vertical design elements in Ljubićeva Street are fragmented into wall masses of different widths, deeply recessed strips of French windows with single-wing and double-wing doors, and walls that freely protrude into the space and frame loggias. In achieving the sculptural quality that Kovačić aspired to in designing the building's exterior, this kind of fragmentation and accentuated play of light and shadow represent a step forward when compared to Lokve (Figure 5).

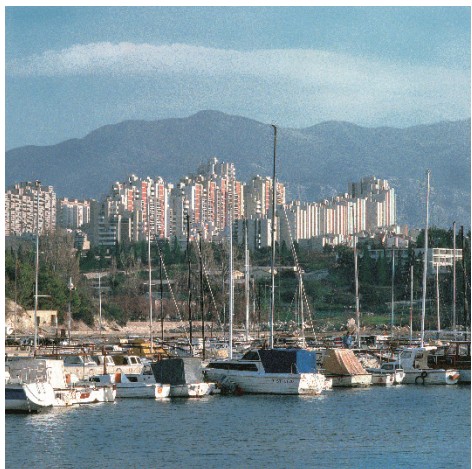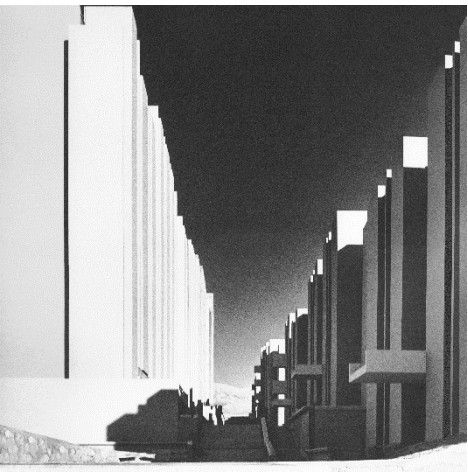

**Figure 5.** View from the sea on Šime Ljubića Street and Dinka Šimunovića Street (**left**), detail of Šime Ljubića Street (**right**). Source: Dinko Kovačić's archive, unknown author.

An important detail in the design of the exterior of the building are railings made up of S-shaped steel bars, which are reminiscent of the stone balusters of baroque palaces. This softness of form against the solid mass of the building's concrete body brings a "certain loveliness" (the architect's expression) (Figure 6).

At first glance, the chosen design of the railing is surprising since it is in complete contrast to the rational impression of the exterior. However, looking at Kovačić's entire oeuvre, there have always been design elements that have given charm to his architecture and have imprinted his unique signature. Kovačić's architecture is recognisable by these small details with which the architect creates the atmosphere.

In front of the apartments on the ground floor, gardens are designed as a space in-between the street and the private area of the apartment. The entrances into the low-rise buildings, which extend along the south side of the pedestrian street, are accentuated by concrete canopies. In the case of the high-rise buildings, the entrance is highlighted by side walls that end in a semicircle towards the top, again in contrast to the straight lines of the building.

By composing dynamic vertical elements of different heights and inventive details, Kovačić made a significant qualitative departure from the previous housing constructions in Split based on rational templates and their multiplications. The revival of the Mediterranean way of living, which had been outlined by Slovenian urban planners, was achieved precisely through architecture. This is the greatest contribution of the designers of Split 3's streets, and the contribution of Dinko Kovačić.

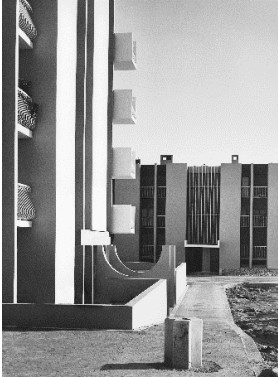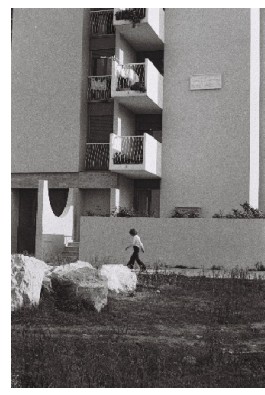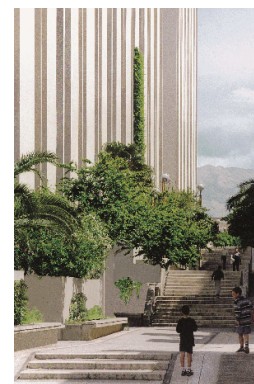

**Figure 6.** Residential building in Šime Ljubića Street, details. Source: Author's archive, photo taken by Vladimir Mušič, 1977 (**middle**) and Dinko Kovačić's archive, unknown author (**left** and **right**).

The apartments are organised to encourage socialising and communication between family members. Even in smaller apartments, Kovačić separated the sleeping area (two to three bedrooms with a bathroom) from the living area. He did this to ensure peace in the intimate part of the apartment. He established a circular link in the living area of the apartment. From the entrance area, it is possible to directly access the living room or the kitchen with the dining room. This can be done by moving from different sides of the centrally placed pantry, toilet, and sometimes the bathroom, in the same block of rooms. Kovačić always tried to design a stage for the events that would take place in the apartment, saying that he had never designed a dining room, but rather lunch. When designing the living rooms of the apartment, he had the contacts of family members in mind, and the "situations in which their eyes meet", the architect pointed out. By doing so, he wanted to fight the alienation "that began to rule with the arrival of empty people who had left all their relationships, friendships, religion, homeland, etc. in the region they came from", explained the architect [Interview with architect Dinko Kovačić, done in Split from 2018 up to 2023.]. In accordance with the position of the apartment in the building and the views of the environment, he arranged the spatial components of a typical apartment differently. This brought a breath of individuality into collective housing. The apartments were designed in accordance with the Guidelines for the Construction of "Split III" [133] and the design programs that were mandatory [134,135], which was a common practice in the design of residential buildings and complexes in former Yugoslavia at that time (Figures 7 and 8).

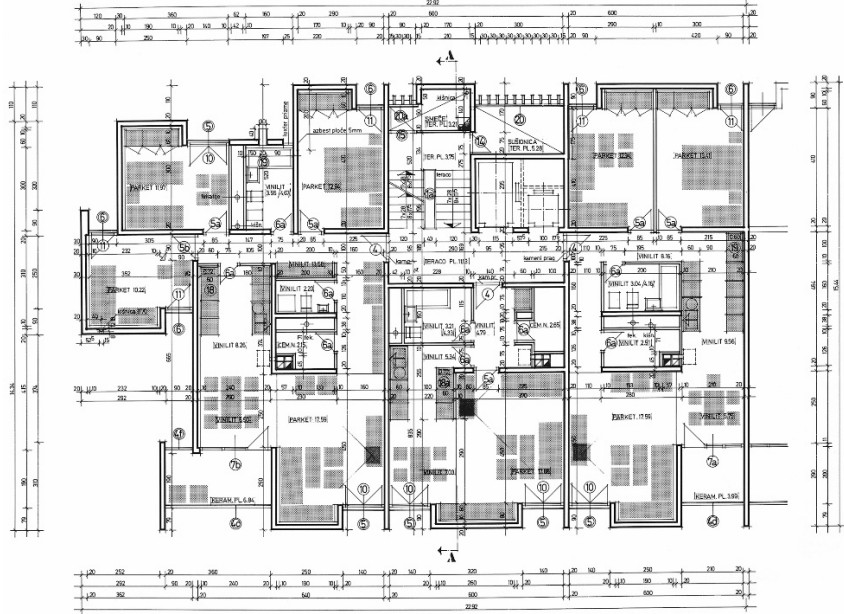

**Figure 7.** *Cont.*

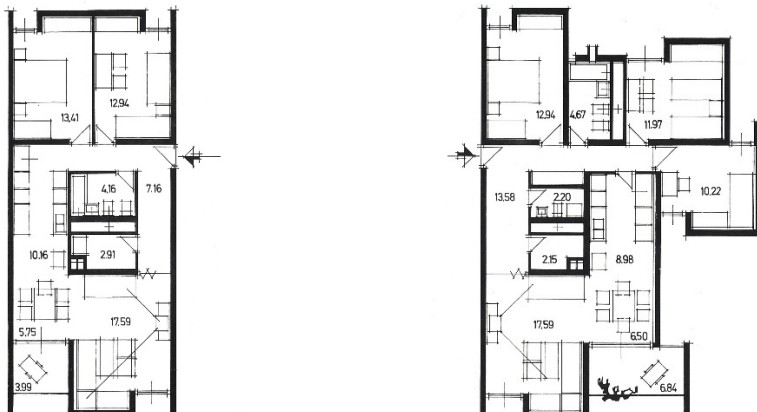

**Figure 7.** High-rise residential building on the north side of Šime Ljubića Street: floor plan of a characteristic floor (**upper**) and floor plans of the apartments. Three-bedroom apartment of 74 m$^2$ (**lower**, **left**) and three-and-a-half bedroom apartment of 90 m$^2$ (**lower**, **right**). Source: Dinko Kovačić's archive, execution design (**upper**), catalogue of apartments of the "I. L. Lavčević" company from Split (**lower**).

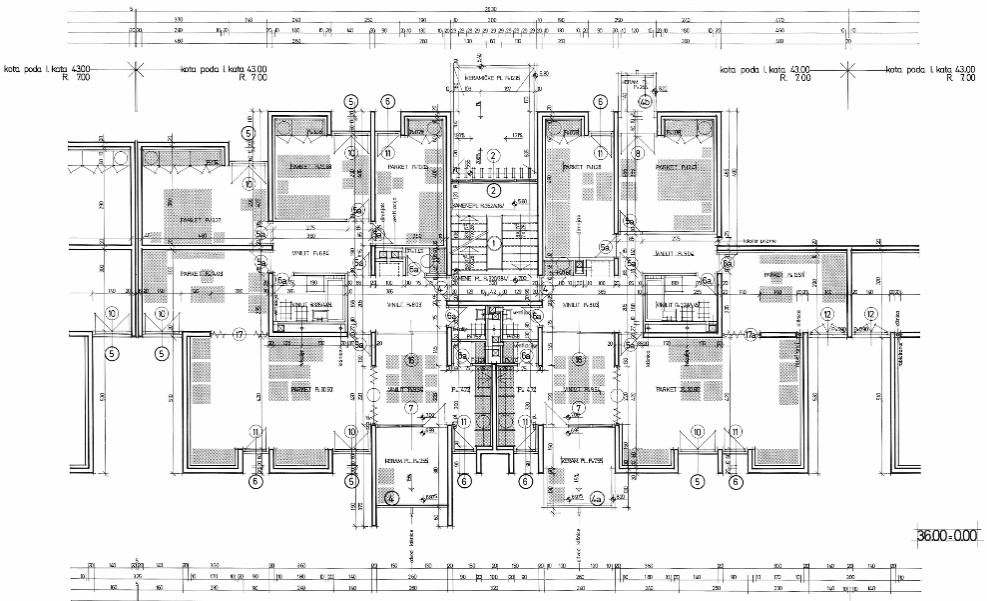

**Figure 8.** Low-rise residential building on the south side of Šime Ljubića Street: floor plan of a characteristic floor. Source: Dinko Kovačić's archive, execution design.

Dinko Kovačić and Mihajlo Zorić won three awards for the Šime Ljubić residential street: the award of the 8th Zagreb Salon in 1973 [109], the state award "Borba" [110], and the annual award "Vladimir Nazor" in 1974 [111]. As stated in the jury's explanation for the "Borba" award, the architects were awarded "for new qualities, new organisation, new approach and insistence on the rational, with the condition that a highly humane expression is equally developed" [57]. Kovačić is proud of the Charter that the Split 3 City District awarded him in 2007, declaring him an honorary citizen of the district: "After 40 years of using the apartments, I saw the expression of friendship of the tenants-users as an extraordinary recognition" [Interview with architect Dinko Kovačić, done in Split from 2018 up to 2023.]. The significance of the charter is even greater because it indicates the tenants' understanding of the importance of modern architecture, which certainly contributes to the preservation and protection of this building as a valuable artefact of modern architecture in Split [Interviews with the tenants of buildings in Šime Ljubića Street in Split 3, done several times in 2021 and 2023.].

Along with the residential Ljubićeva Street, Kovačić also designed other buildings in Trstenik: a series of atrium family houses from 1973–1975 (anagraphic designation: 12a, 12b,..., 14a, 14b Šime Ljubića Street, Split, former Borozan Brothers' Street), residential Dinka Šimunovića Street from 1973–1977 (anagraphic designation: 1–25 Dinka Šimunovića Street, Split), and the "Dalma" department store from 1976 in the same street (anagraphic designation: 10, 12, 14, 16 Dinka Šimunovića Street, Split). For the residential Dinka Šimunovića Street he was awarded the Annual Award of the City of Split for the field of art in 1976 [20,21]. Afterwards, from 1979 until 1982, Odeska Street in the Mertojak area in Split 3 (anagraphic designation: 1–19 Odeska Street, Split) was realised according to Kovačić's design (Figure 9).

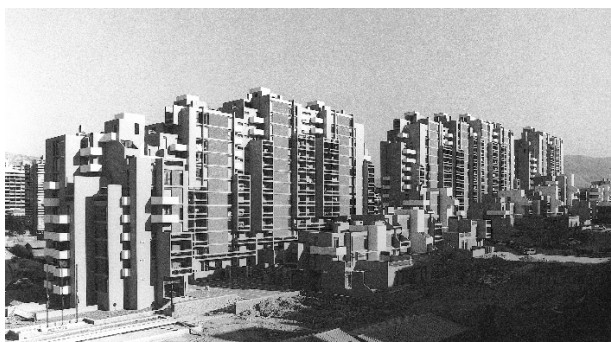

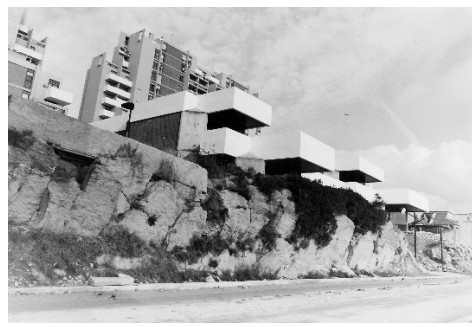
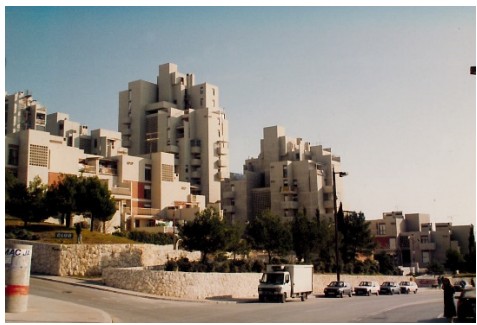

**Figure 9.** Dinka Šimunovića residential street in Trstenik in Split 3 from 1973–1977 (**upper**), *Dalma* supply centre in Dinka Šimunovića Street from 1976 (**lower, left**), and Odeska Street in Mertojak in Split 3 from 1979–1982 (**lower, right**). Source: Dinko Kovačić's archive, unknown author.

### 4.2. Marjanović-Popović Semi-Detached House in Meje, Split

The first family house designed by Kovačić is the semi-detached Marjanović-Popović family house (anagraphic designation: 7, 9 Drvenička Street, Split) built in 1972 in Meje, Split.

The house intrigued many architects; most of the family houses built at that time were executed without any project documentation. This house was an exception and therefore draw attention of many architects. Respected critic of architecture Antoaneta Pasinović, in an article in which she affirmatively writes about the newly realised house, claims that "an endless sea of so-called wild construction has literally eaten the lion's share of the coast" [136]. Architect Ivan Martinac commented on the cultural and social climate in a slightly ironic tone: "Despite the specific cultural climate, which initiated a kind of "direction" in architecture or the so-called "private" design, in which each tenant "knows" best what kind of exterior and interior he needs, the building designed by architect Dinko Kovačić has finally been built in Split. It is the only one that does not have to be ashamed of its southern neighbour: the beautiful Meštrović Castle (author's note: a fortified agricultural estate of the Capogrosso family from the 16th century, owned by the sculptor Ivan Meštrović until 1955)" [137].

Another reason for the great interest from fellow architects in the house was its design. By reinterpreting traditional forms, Kovačić introduced the spirit of Dalmatia into a

contemporary building. The house indeed belongs to the period in which it was constructed, but a repertoire of design elements ties it to the Dalmatian background. Speaking about the Marjanović-Popović house, Antoaneta Pasinović warns that "there is only a small step dividing regionalism from eclectic architectural forms. The distance is so small even the slightest flicker can disrupt everything: all the harmony and compatibility of forms" [136].

The house was designed using prominent, autonomous volumes with deep loggias oriented to the south, east, and west. It was also adapted to the characteristics of the location, and especially to the beautiful views of the sea. The size of these volumes corresponds to the size of traditional Dalmatian houses. They are covered with a single slopping roof whose surfaces point in different directions. By doing so, the architect accomplished the individuality of housing and evoked the memory of old Dalmatian houses. Nurturing his own regional expression, Kovačić continued to develop and upgrade the design of a contemporary house. In many of his subsequent projects, the language of modern architecture is skilfully complemented with elements of traditional Dalmatian architecture. About the design, Kovačić briefly said: "I have tried to give the house a modern feeling by using traditional elements. The memory is present only as a glimpse" [Interview with architect Dinko Kovačić, done in Split from 2018 up to 2023.] (Figure 10).

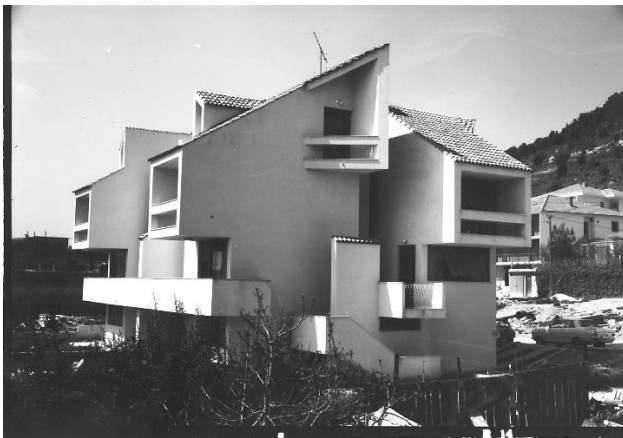

**Figure 10.** Marjanović-Popović semi-detached house, view from the southeast. Source: Dinko Kovačić's archive, unknown author.

Following the example of the Marjanović-Popović house, many similar houses were built in the same neighbourhood and in the vicinity of Split, including hundreds of houses in Kaštela. This demonstrates its successful design and acceptance by his peers and the public. The house in Meje once brought freshness to the established construction practice in Split, which architect Robert Plejić described in an article published in the magazine "Čovjek i prostor" [138].

The house has three floors and an attic. The central volume of the house (first and second floor) is occupied by two-storey apartments for the Popović (eastern part) and Marjanović (western part) families. The purpose of the part of the ground floor and of the attic differs since they belong to different owners. This is because Kovačić respected the special requirements of each family. For example, in the eastern part, which belongs to the Popović family, an apartment was designed, both on the lowest floor and in the attic, because the investor wanted to provide apartments for his then minor sons. Davor Marjanović, the investor of the western part of the house, wanted a smaller one-room apartment for his parents on the ground floor and several storage rooms in the attic instead of the apartment that Popović wanted [Interview with architect Dinko Kovačić, done in Split from 2018 up to 2023 and Davor Marjanović, owner of an apartment in the semi-detached house at 9 Drvenička Street, Split, done in Split on September 18, 2021.] (Figure 11).

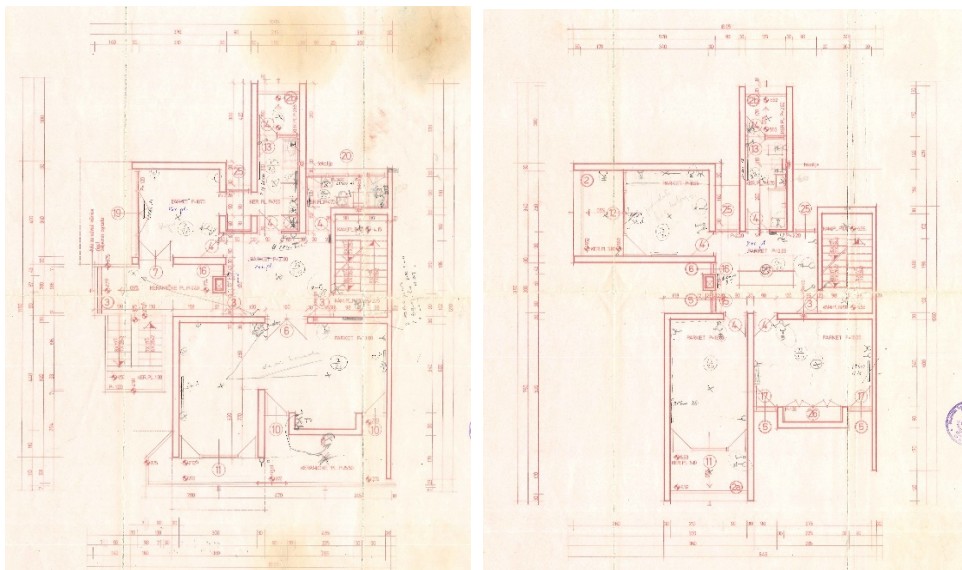

**Figure 11.** Marjanović-Popović family house, ground plan of the first floor (**left**). The staircase, on the left on the floor plan, serves as an exit to the yard of the house, and the one on the right is internal and connects the lower and the upper levels of the two-storey apartment. Ground plan of the second floor (**right**). Source: Author's archive, execution project of the house.

The floor plan of the two-storey apartment is organised in such a way that all the rooms on each floor can be accessed from the larger, centrally placed space. This is the entrance area of the lower level of the apartment and the dressing room on the upper level. This way, the hallway is lost, and the central part of the apartment becomes a meeting place for all family members, something that Kovačić always aimed for when designing living spaces. The rooms of the apartment, as in the floor plan, act as independent units that protrude quite a lot into the exterior space. They lean against the central part of the apartment, to which they are connected by a narrow wall that is sometimes only as wide as the room door. The adjacent rooms are rotated by 90°, so the views from the rooms are directed to different sides accordingly. The design of the indented, moving floor plan brings dynamism to the entire space.

*4.3. Bretanide Hotel in Bol on Brač*

The intensive development of tourism infrastructure on the eastern Adriatic began in the mid-1960s. From 1962 to 1966, systematic valorisation of the Adriatic area was carried out in cooperation with the United Nations Development Program (UNDP). As a result, two important regional spatial plans were carried out; from 1966 to 1969, the Regional spatial plan of the southern Adriatic, and in 1972, the Regional Spatial Plan of the Southern Adriatic of the Upper Adriatic. After these regional spatial plans were released, a series of detailed urban plans were adopted for areas suitable for tourism development, most often chosen because of their natural beauty. Such tourist areas were regularly located in the vicinity of picturesque historic Dalmatian towns, but they were detached from them so that the hotel infrastructure would not impose itself on the natural and architectural heritage and, thus, disrupt the centuries-old harmony.

Due to the organisational difficulty of construction, tourism started to develop somewhat later on the islands than in the coastal areas. In Bol, a town on the island of Brač, the first hotels, "Elaphusa" and "Borak", were built only in the early 1970s. They were designed by architect Žarko Turketo from the Urban Institute of Dalmatia in Split [139]. The hotel area is located to the west of the small Dalmatian town of Bol along the promenade that leads to the pebble-beach cape Zlatni Rat (Golden Horn). The urban design for the hotel complex in Bol was overseen by architect Žarko Turketo [140].

Quite a few tourists were attracted by the beauty of the Golden Horn, so the Municipality of Bol decided to expand the existing hotel facilities, which were not sufficient to accommodate the growing number of guests. In 1982, the municipality announced a competition for the design of a hotel in this exceptional location [141]. Along with architect Dinko Kovačić, three other architects who had extensive experience in designing hotels participated in the competition: Zlatko Ugljen, an architect from Sarajevo, who designed the "Visoko" hotel in Visoko from 1974, the "Bregava" hotel in Stolac, and "The Rose" hotel in Mostar from 1975 [142]; Boris Magaš, from Rijeka, who designed the hotel complex "Solaris" near Šibenik from 1967 and "Haludovo" in Malinska on the island of Krk from 1968 [143]; and architect Ante Rožić from Makarska, who designed many noteworthy tourist buildings, such as hotels "Maestral" from 1965 (designed together with Julije De Luca and Matija Salaj, with the interior done by Bernardo Bernardi) and "Berulia" from 1971, both in Brela, and hotels "Meteor" from 1973 and "Biokovo" from 1979 in Makarska, among others [144].

In a strong competition of architects experienced in hotel design, Kovačić won first prize by offering a new approach to hotel design. He followed his personal view that buildings intended for tourism should satisfy two components: the need for true rest and the need for a parade. In his opinion, hotel construction lies somewhere between these extreme poles.

The "Bretanide" hotel (anagraphic designation: Bol, 50 Put Zlatnog Rata) certainly stands out in Kovačić's architectural oeuvre [145–150]. Designed to accommodate 800 beds, it was built in two phases, in 1984 and 2006. Near the Golden Horn beach, in a dense pine forest and olive groves, the hotel's location required a keen sense of preservation of the found angelic nature, as the architect described it. He carefully placed the buildings on the site and established an agreement concerning the mutual coexistence of concrete and existing vegetation under the motto: "The olive swore to the concrete that it would grow straight" [151] (Figure 12).

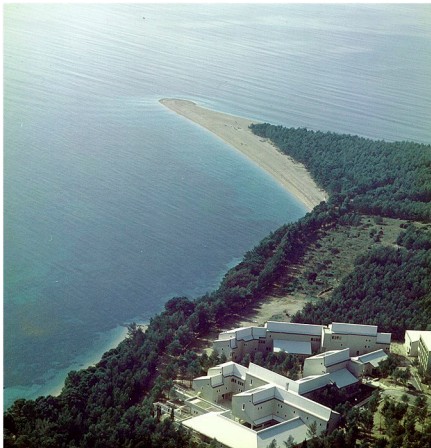 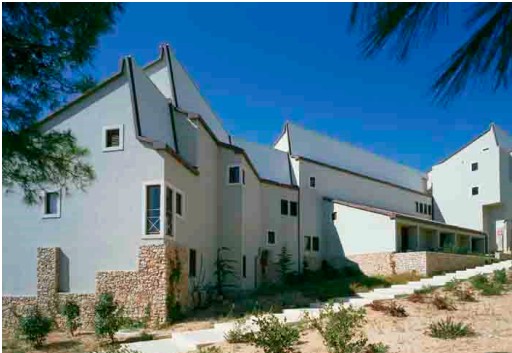

**Figure 12.** *Cont.*

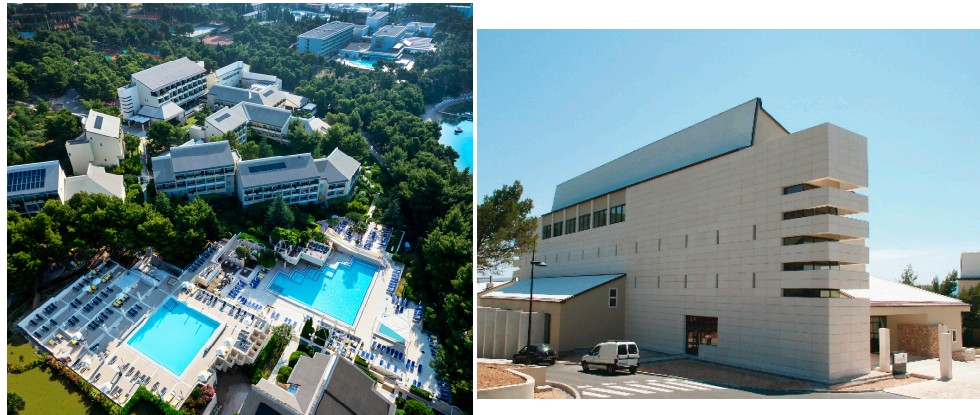

**Figure 12.** "Bretanide" hotel, 1st phase of construction—view of the hotel complex and the Golden Horn pebble-beach cape, (**upper**, **left**) and a detail of the exterior (**upper**, **right**); 2nd phase of construction—aerial view of the hotel complex (**lower**, **left**) and a detail of the exterior (**lower**, **right**). Source: Dinko Kovačić's archive, unknown author.

The floor plan of the hotel draws its origins from the spatial organisation of old Dalmatian towns. Kovačić outlined the small squares, narrow streets, porches, and staircases to achieve the desired atmosphere of outdoor life so characteristic to Dalmatia, as well as other Mediterranean regions. On the square, there is a well top, a sundial "that only counts happy moments", a window for Romeo and Juliet, "na koljeno" shop entrances (finestre inginocchiate - traditional shop doors with a glass cut-out in the wall that serves as a shop window), and also a bifora—a replica from the old library in Bol. To contribute to the warm atmosphere that suits a vacation spot, the architect points out: "Bretanide is made of concrete, stone and diminutives. The cafe is called "Cvitić" (Fleurette), the tavern "Sovica" (Owlet), the tavern "Ptičica" (Birdie), and the souvenir shops "Fjokić" (Tiny Bow)" [33] (p. 274) (Figures 13 and 14).

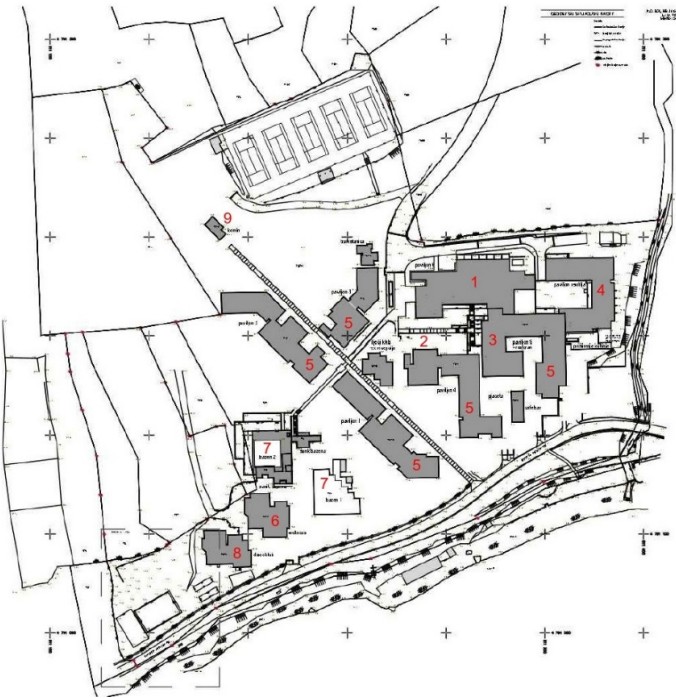

**Figure 13.** *Bretanide* hotel, situational drawing, 1th and 2nd phase of construction, marks: 1—

reception; 2—restaurant; 3—kitchen; 4—staff rooms; 5—guest rooms; 6—wellness; 7—swimming pool; 8—disco-club; 9—"komin" of fireplace with the atmosphere of Vidova Gora. Marks 1–9 added by the author of the article. **Source**: Dinko Kovačić's archive.

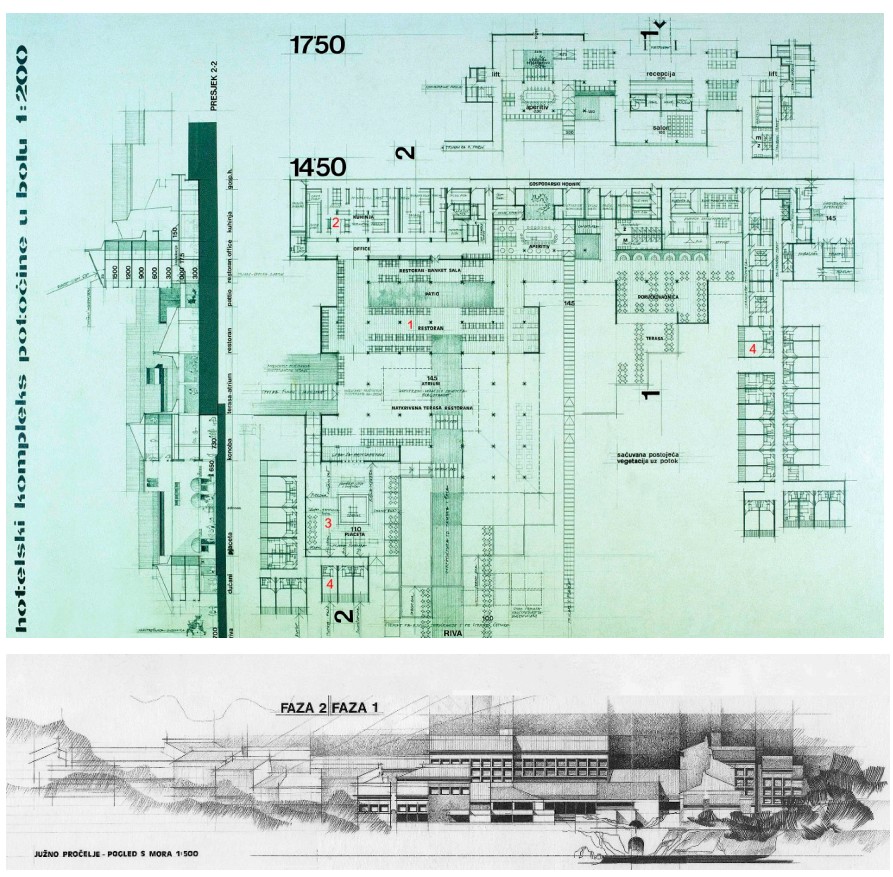

**Figure 14.** *Bretanide* hotel, tender design, floor plan of the ground floor, marks: 1—restaurant; 2—kitchen; 3—Pjaceta square; 4—guest rooms (**upper**) and south façade, view from the sea (**lower**). Sketch by architect Kovačić. Marks 1–4 added by the author of the article. Source: Dinko Kovačić's archive.

The backbone of the hotel complex is a diagonally placed pedestrian path. Along this path, he grouped the hotel facilities and built the autochthonous ambiences of traditional Dalmatian towns.

The streets lead to the sea or to points with distinct ambiences, for example, a fireplace area or "komin" with the feel of Vidova Gora, the highest mountain peak on the island. The prisms of the hotel pavilions are articulated with window openings, loggias, and sloping single-surface roofs with an inventive ridge finish. The sloping roofs covered with white cement panels are reminiscent of the stone roofs of the traditional rural architecture. According to the architect's memory, white stone roofs prevailed in Bol before hotels were built. "In my childhood, when I passed by Brač in a boat, I admired the beautiful whiteness of the stone slabs that covered the houses on the island. That whiteness got engraved in my memory. It was the first thing I noticed in Bol even then, some thirty years later, but only in a few places: most of the houses were already covered with red monk-nun roof tiles. I felt that the roof of 'Bretanide', made of contemporary materials, should be a reminder of those white roofs of Brač" [33] (p. 274). Stone walls partially cover the concrete walls of the ground floor to make the building gently touch the ground. The architect summarises his approach to the harmonious coexistence of the hotel complex and natural environment found at the site by saying: "Only with consistent decency, consistent modernity and, above

all, respect, can even the most subtle demands of both place and time be met" [Interview with architect Dinko Kovačić, done in Split from 2018 up to 2023.].

The second phase of the hotel, planned back in the competition design in 1982, was executed in 2006. At the same time, parts of the hotel dating back from the 1980s were reconstructed. Because of the transition to a market economy, which started the mid-1990s, many hotels built along the Adriatic coast in the second half of the 20th century were significantly adapted, both spatially and functionally. The regulations that prescribe the standards for tourist facilities in Croatia have been continuously changed since the first ones were passed in the 1950s. They have become more demanding, in line with the increase in quality and the addition of new hotel facilities [152]. The existing part of the "Bretanide" hotel passed the process of adaptation to changed requirements. New spaces with a reception, kitchen, and restaurant, as well as new accommodation pavilions, were built. Outdoor swimming pools and sunbathing areas were also added as additional amenities (Figures 15 and 16).

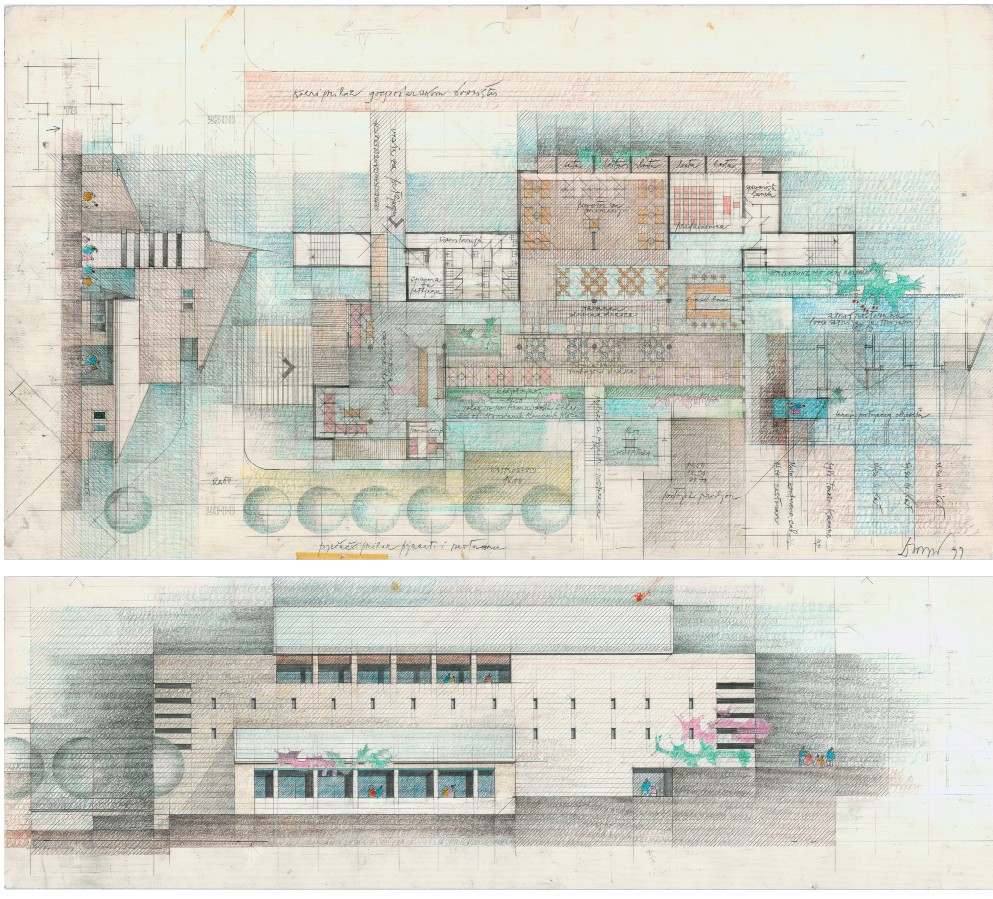

**Figure 15.** "Bretanide" hotel, 2nd phase of construction, floor plan of the ground floor (**upper**) and south façade (**lower**). Sketch by architect Kovačić. Source: Dinko Kovačić's archive, conceptual design.

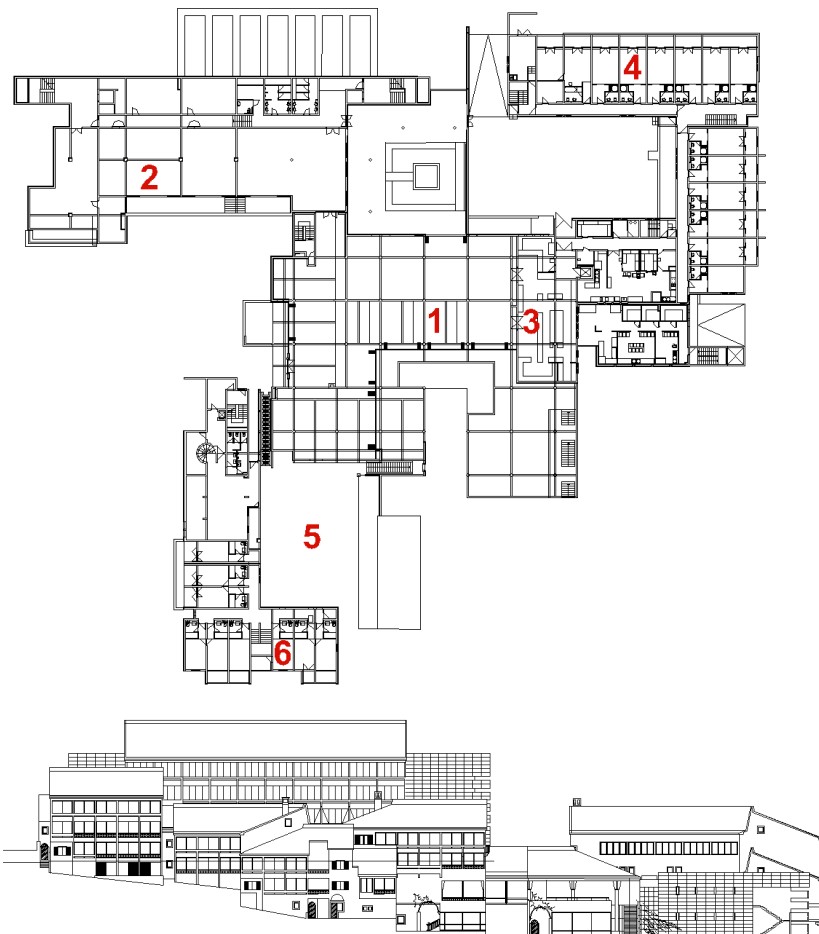

**Figure 16.** "Bretanide" hotel, 1st and 2nd phase of construction, floor plan of the ground floor, marks: 1—restaurant; 2—meeting hall; 3—kitchen; 4—staff rooms; 5—Pjaceta square; 6—guest rooms (**upper**) and south façade (**lower**). Marks 1–6 added by the author of the article. Source: Dinko Kovačić's archive, execution design.

Architect Dinko Kovačić and the owner of the hotel believed that the original distinctive appearance of the hotel should be kept during the reconstruction and addition of new parts. In accordance with this, the southwest walls of the existing pavilions were demolished and new ones, slightly further apart, were built to make the rooms more comfortable. The new, reconstructed roofs were repeated in the same manner as the existing single-surface roofs made of cement panels. The new parts of the hotel from 2006 do not completely match the external appearance of the previously built accommodation pavilions. Kovačić did not seek consistency by copying the forms he applied to the earlier object. Only the roofs remained the same as a link to the older part of the hotel. The facades were shaped in accordance with Kovačić's expressive style. One can notice this in some details on other buildings from the same period, for example, the Faculty of Economics in Split and the Meterize Primary School in Šibenik. With these modifications of the existing part and the construction of new ones, the hotel has a four-star rating.

### 4.4. Secondary School Centre in Split 3

In the 1960s, Split developed into a regional centre. New housing estates were built around the historic centre of the city. Split's construction companies grew stronger and had a constant need to hire construction workers. Good prospects for the development of the city, and thus the construction industry, stimulated the idea of founding a civil engineering school centre where engineers and technicians of the construction and architec-

tural professions—masons, carpenters, and other workers—would be trained. The civil engineering school centre in the Split 3 area was supposed to unite associated facilities in the vicinity: the Civil Engineering Secondary School Centre, the student dormitory for accommodation of school students, the Faculty of Civil Engineering, and the Civil Engineering Institute. The construction of the centre began in the 1970s with the construction of the student dormitory designed by the architect Josip Stubnja in 1969/1970. It was followed by the Faculty of Civil Engineering designed by the architect Kolja Kuzmanić in 1975/1976 and 1978 [12] (pp. 199, 222). In 1979, the architect Dinko Kovačić drew up a design for the Civil Engineering Secondary School Centre but it was not realised at that time.

Towards the end of the 1970s, after the completion of construction of the Split 3 city area and of the sports facilities for the VIII Mediterranean Games held in Split in 1979, the intensity of the city's development decreased. The justification for the completion of the building was questioned because of the reform of secondary school education implemented in 1978 and the lack of funds caused by an economic crisis [153] (pp. 65–68), [Interview with engineer Jakša Miličić, director of the Construction School Center Split from 1965 to 1969, done in Split on 5 May, 2019.].

The school building was not completed by the end of the 1980s when the political tensions in former Yugoslavia intensified, culminating in the Homeland War in 1991. This caused the stagnation of the city and the closure of Split as a regional educational centre for students from nearby Herzegovina.

The Secondary School Centre (anagraphic designation: 11 Matice Hrvatske Street, Split) was built from 1989 to 1992. Following the new needs of the society, the centre accommodated several schools of different educational orientations instead of a single large centre [154,155], [Interview with Sela Tecilazić professor at the School of Design, Graphics and Sustainable Construction, Split, done in Split on March 1, 2019 and Ivan Kovačević headmaster of the School of Design, Graphics and Sustainable Construction, Split, done in Split on March 15, 2019.].

According to the adapted project of the architect Kovačić from 1987, secondary schools specialising in mathematics and information technology, construction, craftsmanship, and chemistry found their home in the Secondary School Centre. "In my opinion, schools belong to the very top of my hierarchical scale. I think it is a success. Four schools function here, and harmony and understanding reign" [Interviews with architect Dinko Kovačić, done in Split from 2018 up to 2023.]. The two-storey school building has 48 classrooms and was designed to accommodate approximately 1200 students in each of the two shifts.

The building was conceived following the urban pattern of a historical Dalmatian town with a square in its centre, which serves as a focal point of social and public life, like Split's "Pjaca" (People's Square). On the square, there are various administrative and public facilities used by all the city's residents, while streets, which branch off from it, lead to residential houses and facilities far from the centre. Kovačić accordingly dedicates the central place in the building to a spacious school square, which is used daily both by students and teachers. Therefore, there are various shared facilities on the square: a library, a coffee bar, four groups of administrative rooms, and four teachers' offices. The square also has a wider public purpose—it is a street in the form of a covered passage that connects parts on the east and west sides of the building. As a result, the school, atypically, has two entrances—east and west. Citizens and students from the nearby faculties of the student campus use this street every day. The architect was consistent in expressing his original idea of a fluid public space by using the design of the stone pavement that connects the interior and exterior spaces. A longitudinal strip of pink stone, Drniš rosalite, stands out in the floor and, in Kovačić's recognisable manner, flows freely into the outdoor spaces. The dashed strip is broken up by yellowish-grey stone: dolit marble from the foothills of Mosor mountain, which was also used to pave the rest of the space (Figure 17).

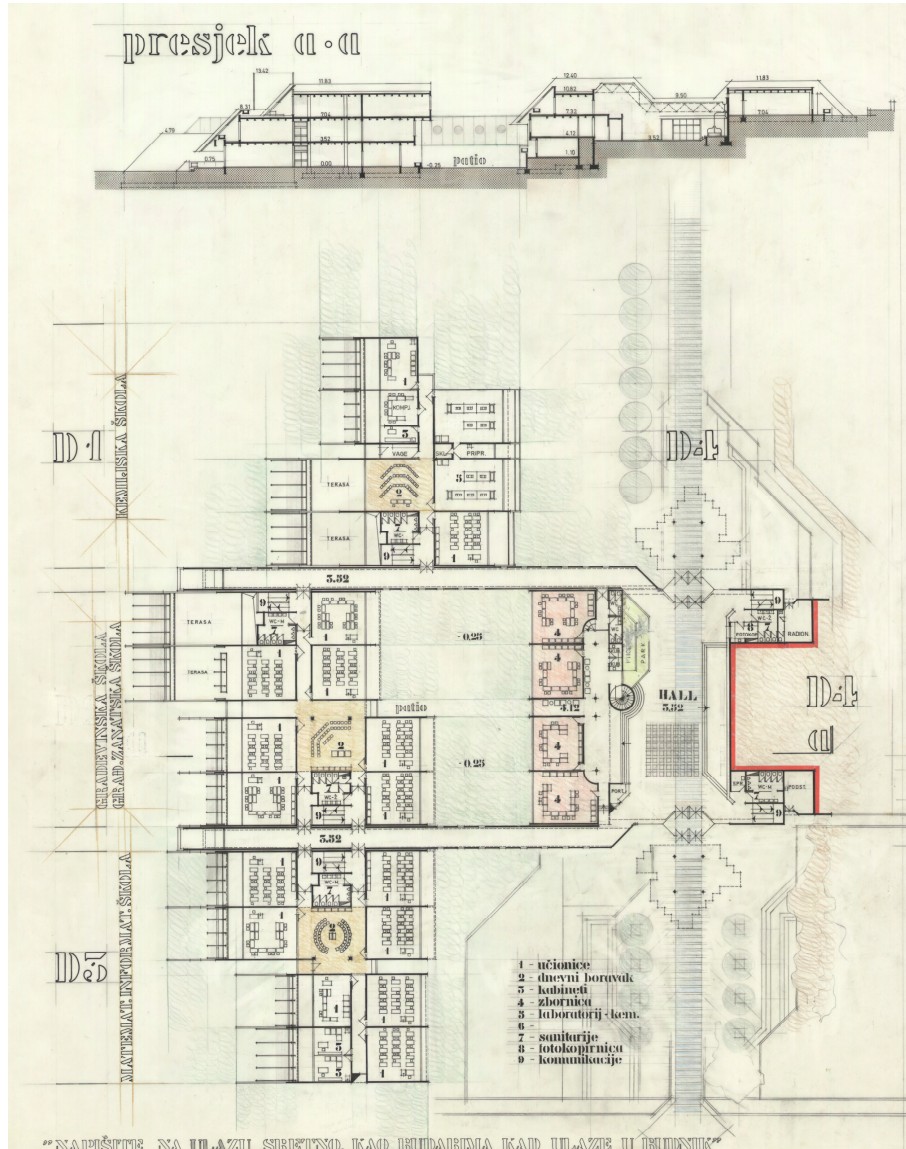

**Figure 17.** Secondary School Centre, floor plan of the ground floor (**lower**) and cross section through of the building (**upper**). Sketch by architect Kovačić. Source: Dinko Kovačić's archive, conceptual design.

The central double-height part of the school was meticulously designed. Daylight passes through the glass roof, which is supported by a metal three-dimensional grid structure. The northern part of this space is raised so that it can serve as a stage for occasional school performances, lectures, and exhibitions. The background of the stage is a wall lined with stone, reminiscent of traditional drywalls. On the opposite side, there is a park with rich greenery and birds, which, with the help of the students, the architect himself takes care of. The park has been decorated with rock brought from the nearby Mosor mountain. The rock spills over the fence wall onto the stone paving of the square. There are also pink benches and streetlamps there (Figure 18).

Two spacious corridors, or "city streets", lead from the square to the school premises in the south. The schools are isolated as "quiet" parts. An inner garden-atrium separates them from the entrance area. The corridors on the ground floor level end with two pavilions outside of the building, in the park on the south side of the school, while on the first floor they end with loggias. Two "streets" on the north side of the school square were also designed. They were planned as access to the school hall, which, however, has not been built. "Schools are like townhouses", points out architect Kovačić, "and they are lined up on the sides of the

street". Instead of an address on the house, here, there is an inscription of a particular school" [Interview with architect Dinko Kovačić, done in Split from 2018 up to 2023.].

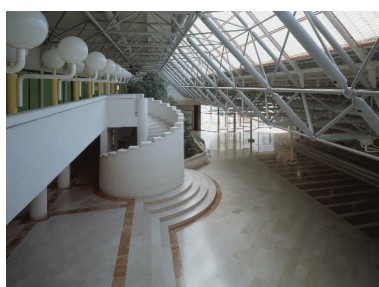 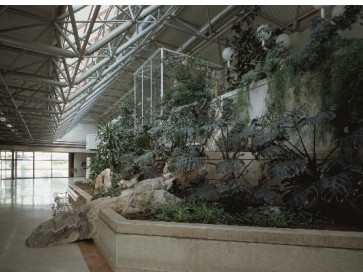

**Figure 18.** Secondary School Centre, the interior view of the school square (**left**) and "park" with greenery, birds, and rocks that spills over onto the paving of the square (**right**). Source: Dinko Kovačić's archive, photo taken by Branko Ostojić.

In preparation for the design of the school, the architect spoke with different experts: teachers, pedagogues, psychologists, and social workers. However, in his opinion, the most useful was a conversation about what a school should be like with the students at a Split high school. One student wrote a message: "Write 'Good luck!' on the door as they do for miners when they enter a mine". This became the motto of the whole project. "It was a serious warning, my task was to confront fear", explained the architect. Apart from the classrooms, he considered it important to provide spaces that "encourage the creation of a good atmosphere, because knowledge and results can only be achieved in a good atmosphere". These spaces intended for students to socialise and rest, which were planned on all floors, were also to be used for preparation, i.e., informal conversations between the teachers and the students about a specific topic so that the students, already interested and prepared, could master the school material with ease and without fear [156,157] [Interview with architect Dinko Kovačić, done in Split from 2018 up to 2023.].

The floor plan of the school was organised by repetition of the classroom as the basic spatial and functional unit. Classrooms that form pairs are separated by wide "city streets" and narrow school corridors. In some areas, the classroom has been omitted, and the architect designed preparation rooms or sanitary facilities in their place. The classrooms are arranged in three levels (ground floor and two additional floors) and are staggered in cross section. This allowed for a stepped finish in the building, with slanted side facades. To bring rhythm to a potentially long façade, the architect shifted groups of classrooms forward on the floor plan.

The exterior design of the building is characterised by the repetition of design elements. Longitudinal load-bearing walls are highlighted on the facade and form the leitmotif of the entire composition. They give the facade a uniform rhythm. The quiet parts with classrooms oriented to the south are protected from the sun by diagonally placed brisoleils. These are made of longitudinal and transverse aluminium strips and are inserted between steal load-bearing profiles. In addition to carrying the brisoleils, these green-coloured profiles split the segment between two load-bearing walls into five smaller longitudinal areas. The architect shapes the city streets, or corridors that penetrate to the outside space and run through the sections with classrooms, in a different way. Kovačić designed corridors to look like the fuselage of an airplane to highlight crowdedness and the movement of students hurrying from the entrance hall to their schools. On the outside, the corridors are covered with green sheet metal and have rounded edges and small round windows like those on an airplane (Figure 19).

Apart from Secondary School Centre in Split 3, Dinko Kovačić designed the Žrnovnica Elementary School (in an area east of Split) built in 1990 and the Faculty of Economics on the Student Campus in Split 3. This was designed and built in two phases: 2000–2001 and 2005–2006. For this building, Kovačić received the annual award "Jure Kaštelan" in the field of culture in 2002, awarded by the "Slobodna Dalmacija" newspaper [114,115].

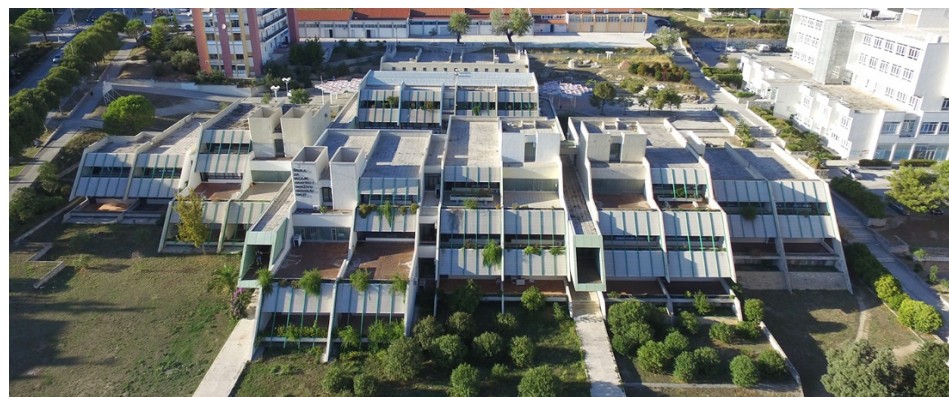

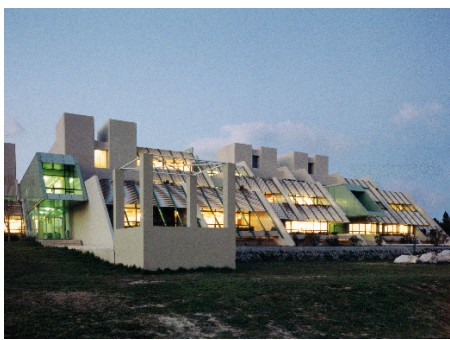 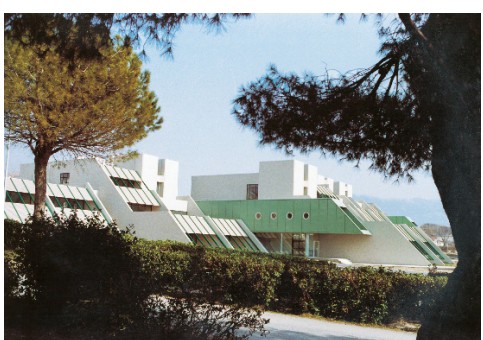

**Figure 19.** Secondary School Centre: aerial view (**upper**); view from the southwest (**lower**, **left**) and from the west (**lower**, **right**). source: Web site of 3rd High School, Split. Available online: https://trema.hr/, accessed on 8 May 2023 (**lower**) and Dinko Kovačić's archive, photo taken by Branko Ostojić (**lower**).

### 4.5. Stupalo Family House in Meje, Split

Kovačić met the new millennium, new social, economic, and political circumstances, and, especially, a new type of investor with two projects: the Commercial Distribution Centre of Tobacco Factory Rovinj in the district of Ravne Njive in Split from 2001 and Stupalo house in Split from 2000.

The Stupalo house (anagraphic designation: 2 Tonča Petrasova Marovića Street, Split) was built in 2000 in Meje in Split [34]. The plot is in the quiet surroundings of the Meje residential district on the southern slopes of Marjan Hill, whose pine forest descends all the way to the sea. There is a beautiful view from the plot. The sea is in the immediate vicinity, at the foot of the access road at the bottom of the plot. On the west side is the villa of the Meštrović family (Meštrović Gallery), which the sculptor Ivan Meštrović built for his family from 1931–1939 according to his own design. The architect stated: "This fact was a determining factor for me, the main thing was to achieve the right measure and design the house with full respect for the neighbourhood" [Interview with Dinko Kovačić, done in Split from 2018 up to 2023.]. For his successfully designed house, which is often called "the most beautiful on the Adriatic", Kovačić received the annual "Drago Galić" Award of the Croatian Chamber of Architects in 2001 for the most successful achievement in the field of residential architecture [113]. In a commemorative article in "Slobodna Dalmacija" on the occasion of receiving the award, Kovačić spoke about what the award meant to him: "Every award is an objectification of quality, but this one is dear to me above all because it comes at the time of the tycoon, profiteering philosophy of 'take as much as possible', for a work that managed to keep measure in everything, especially in the relationship between the investor and me as the architect" [158].

The house of 266 m$^2$ was built on a relatively large building plot of 2100 m$^2$. The house is accessed from the west by a bridge overlaid with wooden beams and covered by a pergola made of rounded steel bars over which bougainvillea hangs, which is reminiscent

of elegant historic walkways to summer houses. This motif is often used in the design of entrances to different buildings, for example, at the Drenova cemetery in Rijeka, the "Hanibal" restaurant in Hvar, and others (Figure 20).

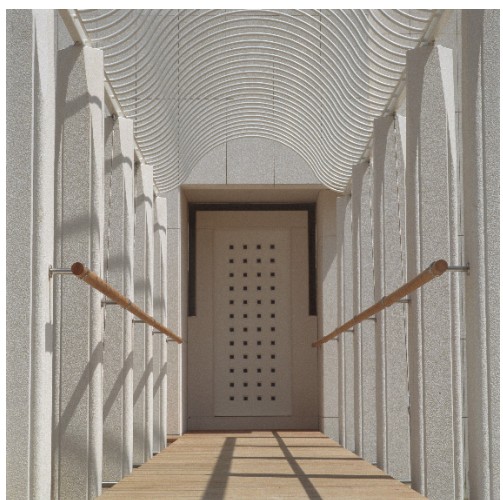

**Figure 20.** Stupalo family house, house entrance. Source: Dinko Kovačić's archive, photo taken by Damir Fabijanić, 2001.

The pedestrian path smoothly transitions, without changing material, into the enclosed space of the house, where wood also predominates on the floors of all rooms, giving warmth to the otherwise modern interior. The hallway passes through the house, dividing the ground floor into two parts intended for different purposes. The southern part is occupied by the living room, which is not level compared to the hallway and other rooms, and the dining room, which is two floors high. The hallway is only separated from the living quarters by a series of load-bearing columns and represents a visual extension of these rooms. In the northern part, the utility section is separated: the kitchen with a pantry, the study, and the garage, which is connected to the apartment by a heated hallway. Two staircases also belong to this portion of the house. One, centrally located and open to the dining room, leads to the bedrooms on the first floor, which have bathrooms and wardrobes attached to them. The second staircase connects the ground floor and the basement of the house (Figure 21).

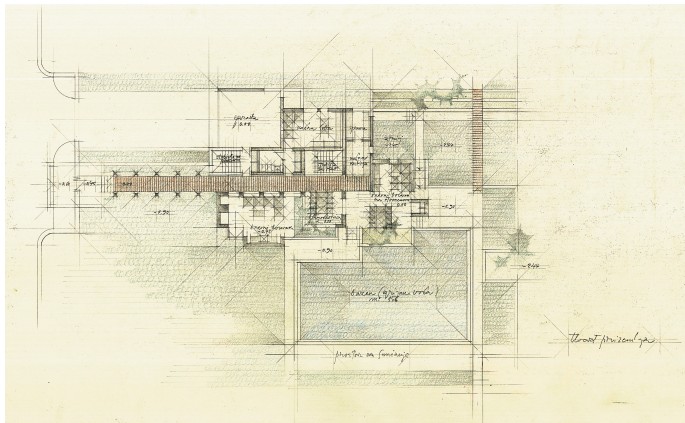

**Figure 21.** *Cont.*

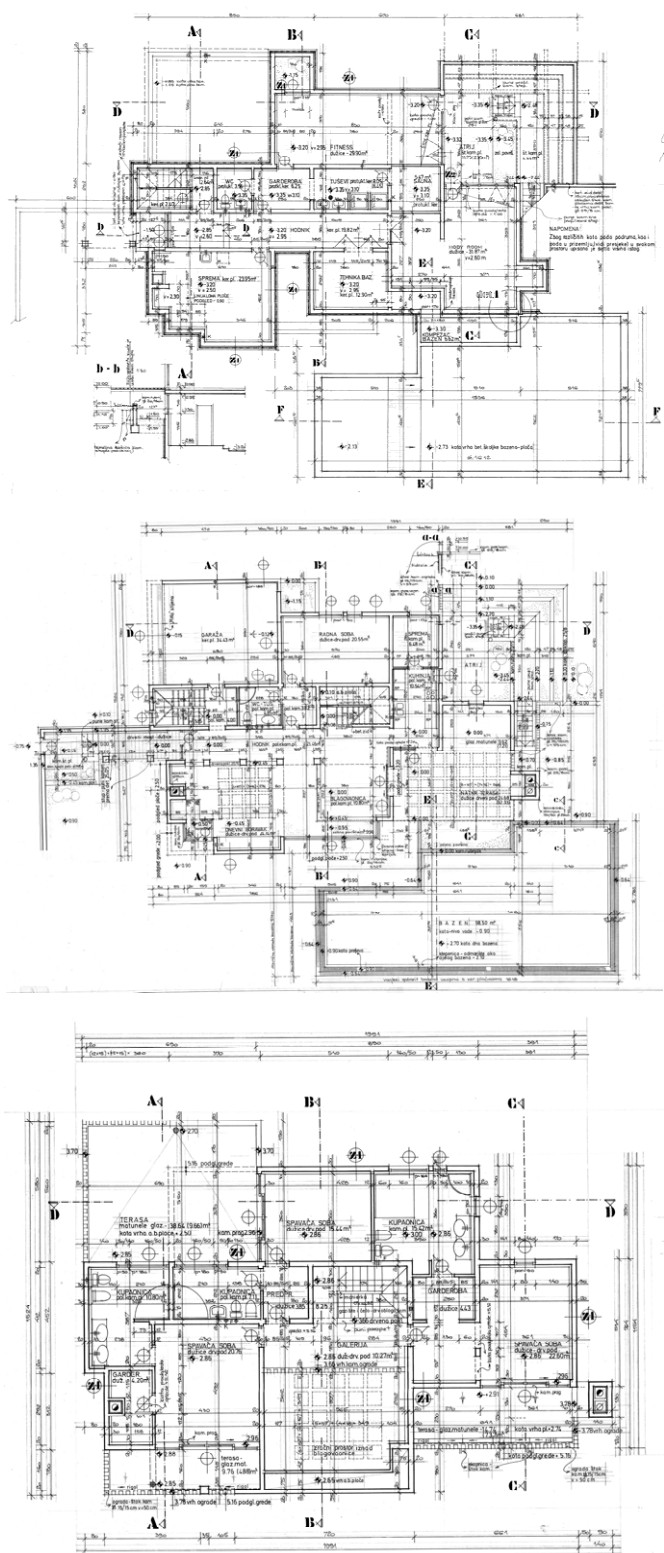

**Figure 21.** Stupalo family house: floor plan of the ground floor, sketch by architect Kovačić (**upper**) and floor plans of the basement (**middle upper**), ground floor (**middle lower**) and first floor (**lower**). Source: Dinko Kovačić's archive, conceptual design (**upper**) and execution design (**middle upper**, **middle lower**, **lower**).

Kovačić paid special attention to the design of the living rooms. The ceiling of the dining room is coffered, and a lighting fixture made of chromed steel pipes that form a

three-dimensional cubic structure hang from it. The floor level of the living room is lowered by 45 cm compared to the dining room and the hallway. By doing so, the architect wanted to create a more intimate atmosphere of the space where the family spend time together. Despite facing south, the living room only has a few openings to the south, so attention is directed towards the fireplace. The parquet floor ends with a strip of white pebbles instead of the usual corner wooden slats along the edges of the room (Figure 22).

On the east side of the house, Kovačić designed an outdoor dining room and an open-air living room. These spaces form a whole with the pool, which, as Kovačić points out, "is not just for swimming, the pool is an important element that participates in building a desired mood". During the day, you can see the sky in the water mirror, and in the evening, the lighting that is reflected from the coffered ceiling of the terrace "actively participates in building the atmosphere of these rooms" [Interview with architect Dinko Kovačić, done in Split from 2018 up to 2023.]. The outdoor terrace–dining room is recessed into the volume of the house and is covered by it. The living room, which is tucked even deeper, provides the tenants with the pleasure of staying outside in the covered, well-ventilated spaces during the summer heat (Figures 21 and 23).

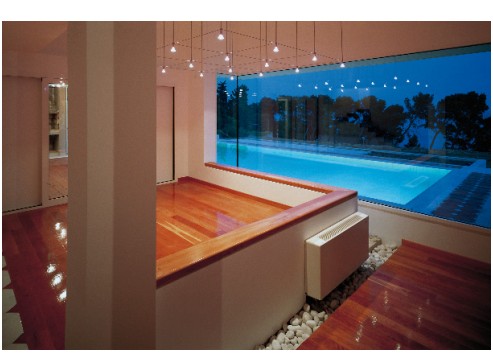
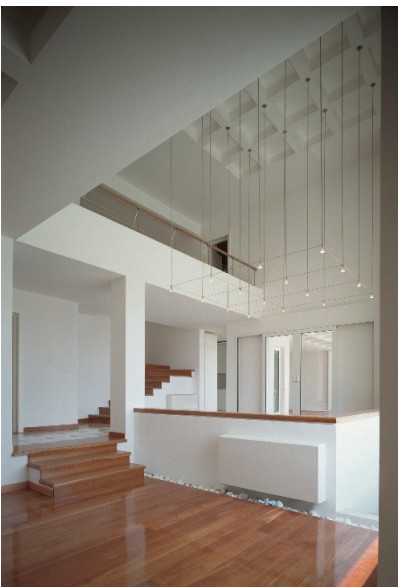

**Figure 22.** Stupalo family house: the interior view from the living room into the dining room and outdoor pool (**left**) and view from the living room into the double-height dining room, hallway, and staircase leading to the first floor (**right**). Source: Dinko Kovačić's archive, photo taken by Damir Fabijanić, 2001.

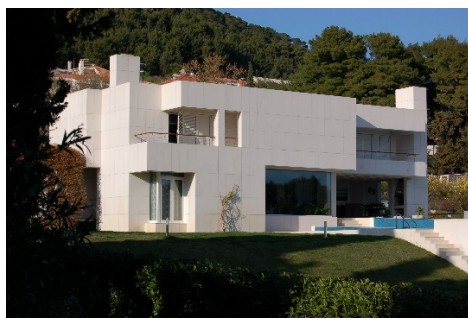
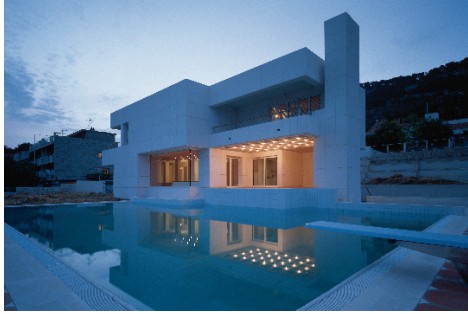

**Figure 23.** Stupalo family house: view from the southwest (**left**) and view from the southeast with the lighting of the outdoor dining room reflecting on the pool (**right**). Source: Author's archive, photo taken by author, 2018 (**left**) and Dinko Kovačić's archive, photo taken by Damir Fabijanić, 2001 (**right**).

The lighting and natural ventilation of the rooms in the basement was addressed with an exit into the inner courtyard. The surroundings cannot be seen from the yard since it is below the level of the terrain around it. Along its perimeter, the architect designed stepped walls between, containing plants. This was done to provide the contents in the basement, which is primarily the exercise room and the hobby room, with both ventilation and a nicer view. He used a similar motif when designing the terrace and the sidewalk portion of the café at the Faculty of Economics in Split, where he ensured a very pleasant stay in an artificial ambient. Similarly, in the "Uvala" hotel in Dubrovnik and the cemetery in Rijeka, Kovačić used skylights and atriums whose walls he covered with stone and greenery to create a feel and view from the inside of a house when there is no natural view (Figure 21).

The building's exterior is shaped by recessing deep loggias in the basic volume and by adding smaller shapes, such as chimneys, to the basic volume of the building. The house is monochrome, covered with stone slabs in a beige tone. This highly aesthetic and, in every detail, very precise and thoughtful work can be associated with the achievements of the architect Richard Meier. Kovačić explained that, in the case of the Stupalo house, he gave up on traditional elements, which he used in many of his buildings by skilfully redesigning and integrating them into the contemporary language of architecture. In his view, a big house such as this one does not tolerate such forms. That is why he designed a house that is rooted in tradition in a different way: this time in Split's modern architecture. "It was the first house in Split that stood out from the earlier ones, in terms of the wishes of the investors, the materials used and the expected luxury", the architect pointed out. With its dimensions, it fits into the environment, which, in the architect's opinion, is of the greatest importance [Interview with architect Dinko Kovačić, done in Split from 2018 up to 2023.].

Following a similar model, Kovačić designed the Skokandić house in Vrbovica Bay near Korčula in 2005, the Vučević house in Varoš in Split in 2007/2008, and the Andabak house in Zagreb in 2010/2011—all significant contributions to Croatian architecture of that time.

### 4.6. Ceremonial Object of the Drenova Central City Cemetery in Rijeka

Although the design process in Kovačić's opinion is a personal and intuitive process, during his many years of practice, he did realise a few projects with his colleagues, e.g., the residential complex in Šime Ljubića Street with Mihajlo Zorić. In co-authorship with Vjekoslav Ivanišević, Kovačić won first prize in the competition for the ceremonial object of the Drenova City Cemetery in Rijeka.

Trsat and Kozala, the old cemeteries in Rijeka, were once isolated places of silence and tranquillity; however, they eventually became part of the city's urban fabric, when the surrounding construction reached their boundaries. With the expansion of the city, the peace of the cemetery was disturbed, but so was the possibility of expanding the cemetery. While keeping the existing cemeteries, the city government chose a new location for the cemetery on the northwest side of the city on a spacious, undeveloped area in the city's outskirts to ensure the necessary space for expansion. The location in Drenova area, a hilly plateau on steep terrain, was envisioned by the General Urban Plan of Rijeka, accepted in 1974. A landscape-type cemetery was planned on a plot of land measuring 44.5 ha [159,160].

The Drenova City Cemetery (anagraphic designation: 2/A Braće Hlača Street, Rijeka) was opened in 1988. Since then, it has been gradually arranged as a garden in which the tombs are carefully divided into groups. These are separated by spacious lawns and smaller groups of trees, which blend into a harmonious garden-like composition. The design of the cemetery was carried out by IGH Rijeka in 1981 [Interview with Nives Torbarina, director of CGG Drenova, Rijeka, done on May 28, 2018.]. The core of this urban and horticultural solution is a wide pedestrian alley that runs through the central part of the cemetery in a southeast–northwest direction. Following the project of Dinko Kovačić, a ceremonial cemetery building was built in the period from 2005 to 2007 at the southeast end of the alley, on a gentle incline with a wide view of the sea. The narrower perimeter around the object was determined by the General Urban Plan of Rijeka from 1974 and the Detailed Urban Plan of the Cemetery prepared in accordance with the said GUP [159,160]. A building of a

specific shape dominates that part of the cemetery, and its pyramidal shape can be seen when looking at Rijeka from the sea (Figure 24).

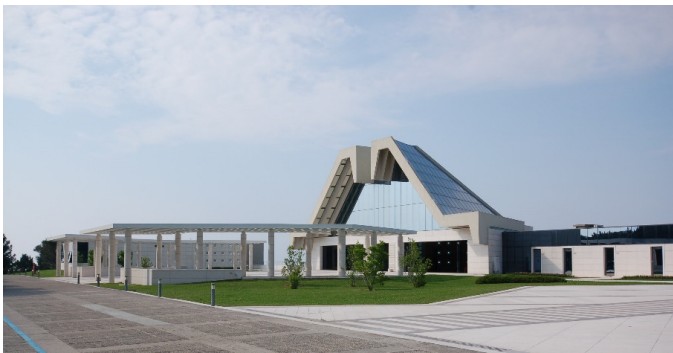

**Figure 24.** Drenova Cemetery in Rijeka: ceremonial object, view from the north. Source: author's archive, photo taken by the author 2018.

Architects rarely contemplate cemeteries in their practice. The topic is very sensitive, and finding an appropriate artistic expression is complex. Therefore, the construction of the central object with a crematorium was preceded by a tender announced in 1981 for the development of its urbanistic–architectural conceptual design. Along with Dinko Kovačić, architects Boris Magaš from Rijeka and Branko Silađin and Berislav Šerbetić from Zagreb were also invited. Another 16 architects also responded to the tender, and, in the end, 20 designs in total were submitted [159,160]. The proposal by Dinko Kovačić and Vjekoslav Ivanišević was rated the best, and the architects received first prize. Architect Zdenko Kolacio, a member of the judging panel, pointed out: "The authors of the first-prized work, Vjekoslav Ivanišević and Dinko Kovačić, decided to emphasise the basic purpose of the building using appropriate forms and to give the ceremonial plateau a visible significance in the landscape of the cemetery. Their solution was highly rated, especially since the architecture provided (and everything else, of course) was in line with the urbanistic, that is, spatial concept of the cemetery area, as well as of that part of the city. It is an example of monumental architecture but of one which blends with its surrounding. Everything was built so that each funeral ceremony preserves its individuality and special meaning, its special sadness, same but also so different from all others" [159] (Figure 25).

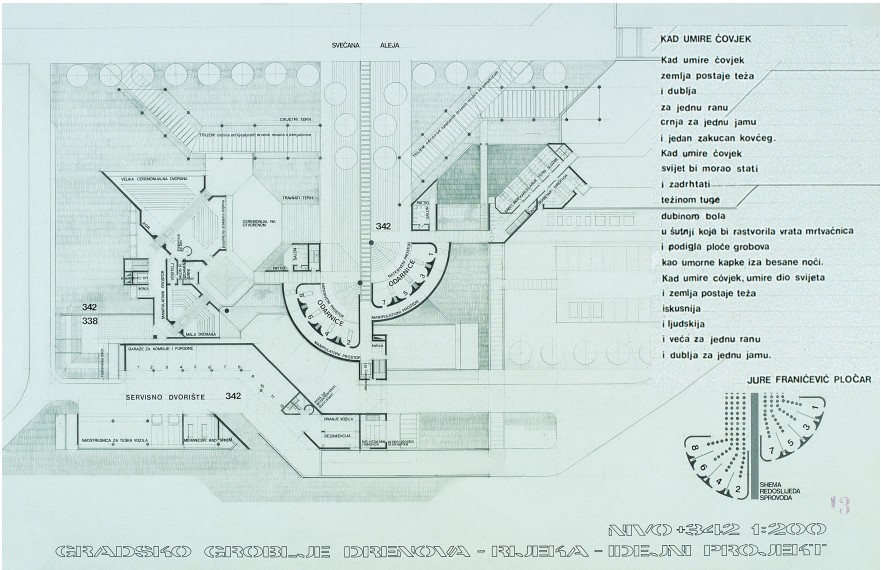

**Figure 25.** Drenova Cemetery in Rijeka: ceremonial object, floor plan. Source: Dinko Kovačić's archive, competiton project.

The construction of the object did not start immediately after the tender, but, according to Dinko Kovačić's testimony, he was called to execute the building project about 15 years later [Interview with architect Dinko Kovačić, done in Split from 2018 up to 2023.]. With the passage of time, he identified certain shortcomings of the awarded work, which stemmed from the requirements of the competition program. Kovačić believed that the proposed eight centrally placed rooms with their catafalques could not provide the necessary intimacy for the send-off and that larger halls intended for commemorations were unnecessary in such complexes. Having convinced the investors of the correctness of his views, instead of elaborating the work for the tender, Kovačić, as an independent architect this time, began the creation of a completely new project. He thoroughly changed the functional scheme and disposition of the building's elements (Figure 26).

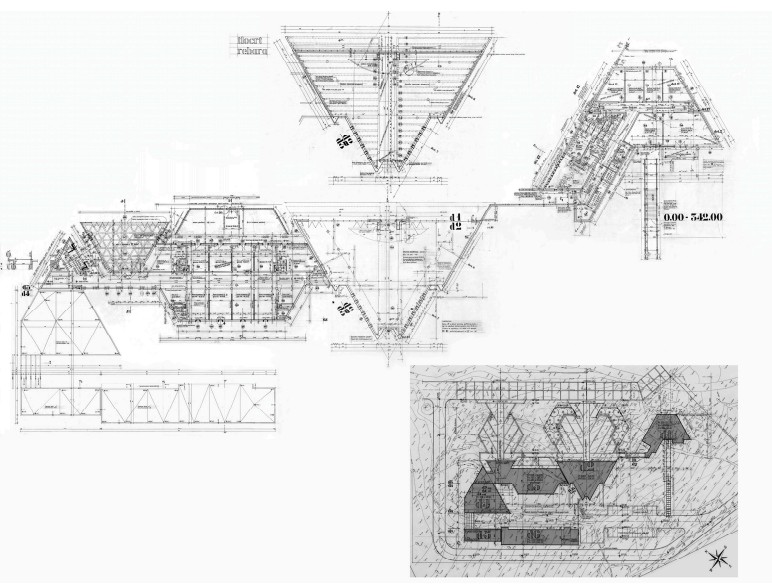

**Figure 26.** Drenova Cemetery in Rijeka: ceremonial object, floor plan (**upper**), and situational drawing (**lower**, **right**). Source: Dinko Kovačić's archive, execution project.

The clear and pragmatic organisation of space results from satisfying the building's function. The result is the division of the elements of this complex into four functional units. These are the ceremonial halls—a large and a small room with catafalques—the administrative part of the complex, and the utility rooms and a crematorium with access to the common utility yard. It will be completed when the capacities of the Zagreb crematorium Urn Grove at the Mirogoj cemetery, which was built in 1985, and the Osijek crematorium, which was built in 2021, become insufficient for the needs of the Republic of Croatia.

Internal corridors, stairs, and an elevator connect the parts into a single whole. Most rooms can be accessed directly from the outside at the level of the upper or lower ground floor of the building.

Kovačić paid more attention to the design of the entrance to the central object of the complex—the large ceremonial hall. It is accessed from the square, which is spatially separated from the pedestrian alley of the cemetery by an airy canopy with a soft structure.

Next to the large catafalque room, another one was placed, which is shaped in the same manner, but has smaller dimensions.

Along with the ceremonial function that the complex required, Kovačić was faced with the great challenge of giving the building a symbolic meaning. "A space for grief—a truly special purpose", Kovačić said about the then recently completed building [48] (p. 20).

The large catafalque room is shaped like a pyramid open towards the sky. The glazed facade that dominates the square is framed by concrete supports, which have been carefully shaped and broken at the top, like hands embracing in a desire to postpone one's parting. The architect achieved a form of strong symbolism and tension; through the gap at the top of the composition, the soul finds its way to heaven (Figure 27).

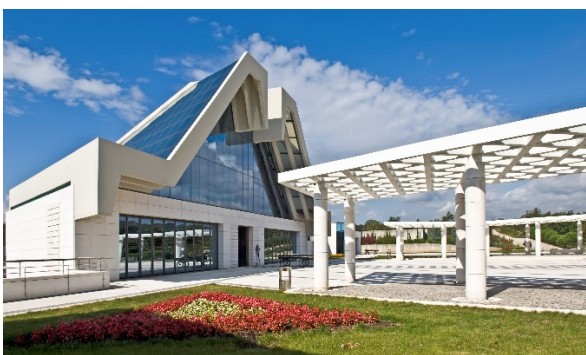

**Figure 27.** Drenova Cemetery in Rijeka: ceremonial object, view from the east. Source: Dinko Kovačić's archive, photo taken by Egon Hreljanović, 2007.

The floor plan modelling of the entire complex was subordinated to the 60° angle rule, and all its parts were consistently designed respecting this norm. The floor plan of both catafalque rooms is, consequently, in the shape of an equilateral triangle. Kovačić was convinced that this form would contribute most to the functionality of the object and its symbolism. The height of the catafalque room is unequal and increases evenly towards the glass membrane, which is in contact with the square. Kovačić placed the bier, thus expressing respect for the deceased, in the quietest part of that space, whose dimensions were tailored to suit man's needs. The choice of the triangular floor plan of the catafalque room is also suitable; it is precisely this shape as to invite one to its interior. It establishes the continuous movement of people from one side of the bier to the other during the funeral.

If necessary, the glass wall can be opened wide and the interior space creates a unified whole with the square. The architect believed that eulogies and commemorations can be held in this area.

The interior is solemn and rich, designed using noble materials with different surface treatments, different colours, textures, and structures with which the architect builds the layering of the space, focusing attention on the bier. The three-part catafalque made of a black polished stone monolith is flanked by columns made of the same material. The columns stand out against the backdrop of the warm olive wood-panelled wall. The wall is perforated with turquoise glass prisms, fireflies, as the architect calls them, through which diffuse light comes in. Kovačić felt that the catafalque, placed in the depth of the space, should be accentuated with discreet lighting. Therefore, he placed a window at the end of the triangular shaped catafalque hall to allow natural light to enter but dispersed it in refracted rays by a screen made of glued glass lamellas. The flat glass "sculptures" are meticulously hand-finished; the edges of each lamella were broken manually. Of all the interiors he has designed, Kovačić considers this detail to be one of the most successful because it harmoniously fits into the space as a whole and contributes to the expected mood (Figure 28).

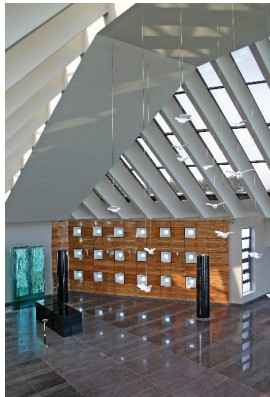 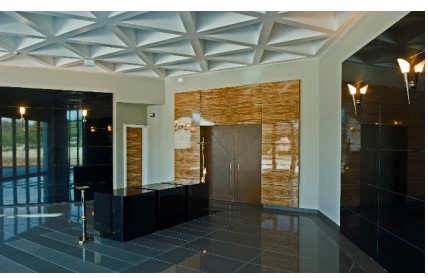

**Figure 28.** Drenova Cemetery in Rijeka: the interior of the large catafalque room (**left**) and of the small catafalque room (**right**). Source: Dinko Kovačić's archive, photo taken by Egon Hreljanović, 2007.

Tinted glass surfaces of blue reflection through which the sky can be seen were inserted between the concrete ribs. These ribs, in turn, shape the pyramidal roof structure and divide it into smaller segments. A flock of white porcelain doves, made by sculptor Vasko Lipovac, flickers in the sky in a solemn bluish atmosphere, bringing joy (Kovačić's expression) to this remarkable space.

He ended his professional career with the realisation of two residential buildings, in Rovinj in 2007 and in Supetar on the island of Brač in 2008. The last project built according to his design is the Meterize Primary School building in Šibenik, in 2010–2012.

## 5. Discussion and Conclusions

Dinko Kovačić is a prominent Croatian architect with an extremely large and high-quality oeuvre. For his architectural realisations, he won many professional awards, including all the most significant statal and institutional awards. He is a respected university professor, who is very popular among students. For his contribution in the field of architecture, he was admitted to the Croatian Academy of Sciences and Arts. He practiced his profession with love and great enthusiasm for about 50 years. He approached each project very professionally, introducing his knowledge and skills at that moment. "And so, in the end, I always have a wonderfully clear conscience that each of my buildings is the best building I could have made at that moment..." [47].

He has lived all his life in his hometown Split in Dalmatia, the Mediterranean part of Croatia. Thus, the Mediterranean heritage and tradition form the fabric of his architectural work.

In the early days of his professional career, he designed buildings of pronounced monochromatic plasticity and well-measured proportions (Ljubićeva Street, Split 3). In Dinka Šimunovića Street in Split 3, the surfaces are additionally articulated with materials of different colours and textures. In both buildings, the play of light and shadow towards the sunny sky was achieved by rich plasticity of the volumes, which have indented layouts and consist of fragmented masses.

Regionalism, present in his work from the 1970s, as can be seen in his large residential buildings or more straightforwardly in the family houses, became his main theme in the 1980s. When designing the "Bretanide" hotel in Bol on Brač in 1984 and the school centre in Split 3 in 1989, he conceptually relied on the urban matrix of the old Dalmatian town or the historic core of Split.

The arrival of the new millennium brought the independence of Croatia and the transition from socialism to neoliberalism. When designing houses for new wealthy investors, Kovačić changed his style and adapted it to the new requirements. These are architectural works of refined modernist expression with flat surfaces and pronounced sculptural volumes. Plasticity was achieved with the shadows created by the dramatically protruding balcony surfaces and the loggias and terraces, which are significantly recessed into the building.

The quality of his architecture puts Dinko Kovačić alongside his European architectural contemporaries of the second half of the 20th century.

His houses are expressive, appealing, photogenic, and rich with details. Many architects, art historians, and journalists have written about them in the local daily press and professional publications. Kovačić likes to talk with journalists about his architecture, as well as about the problems in the field of architecture. Although, despite his narrative ability being extremely abundant, Kovačić did not write a lot. His attitudes towards architecture, and his design method, are described in more than 200 articles in the daily press and in professional publications.

Dinko Kovačić began his architectural practice at a time when the importance of the connection with the historical context and the role of images and symbols was becoming stronger in architecture. At that time, as an answer to the impoverished and reduced architectural language of the international style, postmodernism emerged, with its scenographic, nontectonic use of historical elements. Nevertheless, some architects across the world concurrently bound the meaning of place to the materiality of their architectural

work successfully. This reinterpretation of the cultural context expressed through modern technological means was recognised as a common thread of critical regionalism. That was the basis for Dinko Kovačić's specific work.

Deeply rooted in the Mediterranean context, Kovačić conveys the forms of traditional Dalmatian architecture into contemporary expression. Apart from individual elements, such as stone walls, sloped roofs, stone-paved footpaths, he uses the pattern of the historic Dalmatian towns with their narrow streets, small piazzas, and hidden gardens as a basis for the spatial organisation of his houses.

Beyond this visual attraction, which is without a doubt the sign of the time in which it was created, there is a more subtle timeless and universal design approach. Juhani Pallasmaa, his contemporary from geographically and culturally distant northern Europe, notes: "Modernism at large—its theory, education as well as practice—has focused more on form and aesthetic criteria, than the interaction between the built form and life, especially mental life" [161] (pp. 4–19). This is precisely where Dinko Kovačić is well ahead of his time. His main goal was to create architectural space that is a precondition for happy living, as his gift to other people. When designing apartments or family houses, his main concern was to create a space where the eyes of the tenants meet. The hotel, in his words, is somewhere between the extreme poles of the place for quiet rest and a place for a parade. His hotel "Bretanide" on the island of Brač exactly outlines these two poles, offering an architectural background for a quiet rest in a Mediterranean garden or a background for a parade in stone-paved promenades and piazzas. From his point of view, the prime mission of the school building is to fight against fear and to create a good atmosphere as a vital precondition for gaining knowledge. Thus, the hall of his Secondary School Centre in Split speaks to this. Even in the condition of the neoliberalism of the new millennium, his luxurious Stupalo house still fits into the environment with its dimensions and reduced architectural language, without any exaggeration in its appearance. Being in fact modest in its language despite its luxury, it shows the architect's attitude, in which achieving measure and agreement are the most important tasks of an architect.

When designing his buildings, Dinko Kovačić primarily creates the experience of architectural space. Once asked in one of his lectures which materials his houses were made of, Kovačić answered: "Of glass, stone and concrete", and then continued: "of young men and women. The smile is also a material that I build a house with" [71].

Pallasmaa distinguishes "two qualitative levels of imagination; one that projects formal and geometric images while another one simulates the actual sensory, emotive and mental encounter with the projected entity. The first category of imagination projects the material object in isolation, the second presents it as a lived and experienced reality in our life world. ( . . . ) The lived characteristics—the building as a setting for activities and interactions—call for a multi-sensory and empathic imagination" [161] (pp. 4–19). Mere spatial perfection is never the only target of Kovačić's architecture, although he thoroughly masters the process of materialisation of his ideas. Dinko Kovačić in his architecture always anticipates the experience of the future users. His design method is most clearly described in his advice to the students of architecture, by saying: "Do not draw a dining room, draw a lunch" [71].

**Author Contributions:** Conceptualization, V.P.J. and N.M.K.; methodology, V.P.J. and N.M.K.; validation, V.P.J. and N.M.K.; formal analysis, V.P.J. and N.M.K.; investigation, V.P.J. and N.M.K.; resources, V.P.J. and N.M.K.; data curation, V.P.J. and N.M.K.; writing—original draft preparation, V.P.J. and N.M.K.; writing—review and editing, V.P.J. and N.M.K.; visualization, V.P.J. and N.M.K.; supervision, V.P.J. and N.M.K.; project administration, V.P.J. and N.M.K.; funding acquisition, V.P.J. and N.M.K. All authors have read and agreed to the published version of the manuscript.

**Funding:** This research was funded by University of Split, Faculty of Civil Engineering, Architecture and Geodesy, project KK.01.1.1.02.0027, which is co-financed by Croatian Government and the European Union through the European Regional Development Fund—the Competitiveness and Cohesion Operational Programme and University of Zagreb, Faculty of Architecture, institutional project "Conservation of the Twentieth-Century Architectural Heritage in Principles and Practice".

**Data Availability Statement:** Not applicable.

**Conflicts of Interest:** The authors declare no conflict of interest. The funders had no role in the design of the study; in the collection, analyses or interpretation of data; in the writing of the manuscript; or in the decision to publish the results.

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
