# Peer review of "Specific Design Approach of Croatian Architect Dinko Kovačić: The Coexistence of Modernism and Tradition in the Second Half of the 20th Century"

_heritage, doi:10.3390/heritage6070265_

Round 1
Reviewer 1 Report
The article is a relevant reaction since the quality of architectural works by east European architects is progressively disclosed and becomes part of the international discourse, particularly in the field of modern architecture. However, I believe there are two issues that are impeding this piece of writing. First, consider the length and organisation of the document. The article is now split into two sections, the first of which tells the biography of the architect in question, and the second of which discusses some of his larger-scale projects. The biography section is very lengthy and detailed. This section could be shortened and simplified by emphasising the pivotal period in the architect's career that led to his modernist practises. This will allow the writers to analyse the project in question more critically by comparing and contrasting. The paper might be restructured in this fashion. Instead of examining the projects one by one, the writers may emphasise certain characteristics of Dinko's work and philosophy when discussing the biography, and then examine the projects in light of that. This would bring more rigour to the conclusion debate. Second, the paper's disciplinary aspects. At this point, the article seems to be a good match for a publication that discusses architectural history and philosophy. It is unclear how this work linked with the notion of heritage and addressed the journal's major areas of focus. The writers must address this explicitly.
Overall, I believe the work need a significant rewrite to address the flaws raised above.
The quality of langauage is fine.
Author Response
Dear Reviewer 1,
Thank you for reading our article and for your review. We believe that your suggestions have significantly improved the article.
Point 1
The article is a relevant reaction since the quality of architectural works by east European architects is progressively disclosed and becomes part of the international discourse, particularly in the field of modern architecture.
However, I believe there are two issues that are impeding this piece of writing. First, consider the length and organisation of the document. The article is now split into two sections, the first of which tells the biography of the architect in question, and the second of which discusses some of his larger-scale projects. The biography section is very lengthy and detailed. This section could be shortened and simplified by emphasising the pivotal period in the architect's career that led to his modernist practises.
Response 1: The biography section has been shortened and significantly restructured. The pivotal aspects of the architect’s career have also been emphasised. The added subsection about socio-political context has been suggested by other reviewer.
Point 2
This will allow the writers to analyse the project in question more critically by comparing and contrasting. The paper might be restructured in this fashion. Instead of examining the projects one by one, the writers may emphasise certain characteristics of Dinko's work and philosophy when discussing the biography, and then examine the projects in light of that. This would bring more rigour to the conclusion debate.
Response 2: The biography section has been significantly restuctured. This has caused minor changes in the second part which discusses some of the architects larger-scale projects. The idea of examining projects according to certain characteristics would have been engaging, but could not be carried out because the specific approach of the author appears in all projects presented. The peculiarity of the architects approach is presented in the rewritten conclusion which is now much more precise.
Point 3
Second, the paper's disciplinary aspects. At this point, the article seems to be a good match for a publication that discusses architectural history and philosophy. It is unclear how this work linked with the notion of heritage and addressed the journal's major areas of focus. The writers must address this explicitly.
Response 3: The paper will be processed as a *regular paper*, and not in Special Issue “Preservation and Revitalisation of Built Heritage”.
Point 4
Overall, I believe the work need a significant rewrite to address the flaws raised above.
Response 4: The article has been significantly rewritten. Thank you again for your suggestions for the improvement.
Thank you very much for your time and efforts.
Kind regards,
Authors
Author Response
Respected Reviewer 2,
Thank you for reading our article and for your review. We believe that your suggestions have significantly improved the article.
Point 1
The authors of article have qualitatively researched the architectural work of the architect Dinko Kovačić.
Response 1:
We, the authors, kindly thank you for commending our work.
Point 2
When the authors analysed the architect's architectural work, they very mildly described social circumstances (e.g. “social order” may be better word is socialism) but they did not connect them with economic and political circumstances.
It is recommended to briefly describe that in 1945-1991 socially engaged planning was part of the socialism (they mentioned “social order”). In socially engaged planning (1945-1991), financial resources are intended for social welfare (social order). Large residential, tourist, educational and industrial complexes were planned on the basis of spatial plans (which took people into account) - especially in the 60s and 70s of the last century.
Response 2:
Social circumstances have been described and connected with the economic and political circumstances.
Point 3
In the early 90s (primary and secondary school in Split of Dinko Kovačić) are the legacy of spatial plans from period of socialism?
Response 3:
The Secondary School Centre in Split 3 was built according to a modified and adapted design of the Civil Engineering Secondary School Centre from the socialist period and this has been mentioned in the article.
Point 4
Second recommendation - references: Why are literature sources (references) that are exclusively published in Croatian (not published in English - newspapers, articles, books, etc.) translated directly into English? In this way, the conclusions cannot be verified with Croatian language.
Response 4:
The titles of literature sources (references) which were published in Croatia and in the Croatian language (articles in professional and scientific magazines, newspaper articles and books) have been written in both Croatian and English.
Thank you very much for your time and efforts.
Kind regards,
Authors
Reviewer 3 Report
The paper focuses on the figure of Dinko Kovačić, a prominent Croatian architect and university professor. The work is impressive, well organised and quite clear. The paper well describes the major role played by the architect in Croatia and his large creativity. The bibliography is vast, perhaps a little excessive. However, the conclusions seem a bit naïve to me. They are a sort of synthesis of already known contents but they do not clearly express what is the Influence of Dinko Kovačić today and its surprising contemporaneity. I suggest authors to reconsider re writing this last part.
Minor editing of English language required
Author Response
Respected Reviewer 3,
Thank you for reading our article and for your review. We believe that your suggestions have significantly improved the article.
Point 1
The paper focuses on the figure of Dinko Kovačić, a prominent Croatian architect and university professor.
The work is impressive, well organised and quite clear.
The paper well describes the major role played by the architect in Croatia and his large creativity.
Response 1:
We, the authors, kindly thank you for commending our work.
Point 2
The bibliography is vast, perhaps a little excessive.
Response 2:
The bibliography has been slightly reduced. The reason why it is somewhat vast is because we wanted to present how interesting this architecture was to the public.
Point 3
However, the conclusions seem a bit naïve to me. They are a sort of synthesis of already known contents but they do not clearly express what is the Influence of Dinko Kovačić today and its surprising contemporaneity. I suggest authors to reconsider re writing this last part.
Response 2:
In accordance with your suggestions, the conclusion has been restructured, rewritten and certain pieces of information have also been added.
Thank you very much for your time and efforts.
Kind regards,
Authors
Round 2
Reviewer 1 Report
The authors has responded to the issues raised and I am satisfied with response.